# Quantum collider probes of the fermionic Higgs portal

Ulrich Haisch[1★], Maximilian Ruhdorfer[2†], Konstantin Schmid[3,4‡] and Andreas Weiler[4∘]

**1** Werner-Heisenberg-Institut, Max-Planck-Institut für Physik,
Föhringer Ring 6, 80805 München, Germany
**2** Laboratory for Elementary Particle Physics,
Cornell University, Ithaca, NY 14853, USA
**3** Dipartimento di Fisica e Astronomia "G. Galilei", Università di Padova,
Via Francesco Marzolo 8, 35131 Padova, Italy
**4** Technische Universität München, Physik-Department,
James-Franck-Strasse 1, 85748 Garching, Germany

★ haisch@mpp.mpg.de , † m.ruhdorfer@cornell.edu ,
‡ konstantin.schmid@phd.unipd.it , ∘ andreas.weiler@tum.de

## Abstract

We explore the sensitivity of future hadron colliders to constrain the fermionic Higgs portal, with a focus on scenarios where the new fermions cannot be directly observed in exotic Higgs decays. This portal emerges in various models including twin-Higgs scenarios and dark matter models, posing significant challenges for collider tests. Working in an effective field theory (EFT), we determine the reach of the high-luminosity option of the Large Hadron Collider (HL-LHC), the high-energy upgrade of the LHC (HE-LHC) and a proposed Future Circular Collider (FCC) in probing the fermionic Higgs portal through off-shell and double-Higgs production. Notably, we find that quantum-enhanced indirect probes offer a better sensitivity than other direct Higgs measurements. We argue that this finding is valid in a wide class of ultraviolet realisations of the EFT. Our study presents a roadmap of a multifaceted search strategy for exploring the fermionic Higgs portal at forthcoming hadron machines.



# 1  Introduction

The discovery of a new spin-0 state at the Large Hadron Collider (LHC) by the ATLAS and CMS collaborations [1,2] in 2012 has ushered in a new era in high-energy particle physics. In the last eleven years, it has been established by a concerted experimental effort that the discovered 125 GeV state has approximately the properties of the Higgs boson as predicted by the Standard Model (SM) of particle physics [3,4]. This finding has opened up new avenues in the pursuit of physics beyond the SM (BSM) by performing Higgs precision measurements at the LHC similar to what has been done at LEP and SLD in the case of the $Z$ boson [5].

   Recent important examples of such LHC measurements include the latest constraints on the invisible and unobservable branching ratios of the Higgs boson, which are approximately 10% and 20%, respectively. Indeed, the obtained limits impose stringent restrictions on numerous BSM scenarios featuring either prompt or displaced exotic Higgs decays — see for example [6] for a recent comprehensive review. Specifically, these bounds apply when the masses of the new states are below approximately half of the Higgs-boson mass, around 62.5 GeV. Testing BSM scenarios becomes notably more challenging when dealing with new particles that primarily couple to the Higgs boson and have masses exceeding the kinematic threshold. In fact, there are only two identified categories of collider measurements offering sensitivity to such BSM model realisations. First, the pair-production of new particles in off-shell Higgs processes, including vector-boson fusion (VBF), $t\bar{t}h$, and the gluon-gluon-fusion channel [7–19]. Second, investigations into virtual effects stemming from the exchange of new particles in loops, contributing to processes like associated $Zh$, double-Higgs or off-shell Higgs production [20–27].

   The goal of this work is to explore the sensitivity of future hadron collider measurements in constraining the fermionic Higgs portal

$$\mathcal{L}_{H\psi} = \frac{c_\psi}{f} |H|^2 \bar{\psi}\psi \,, \tag{1}$$

focusing on the case where the new fermions $\psi$ are not accessible in exotic Higgs decays. Here $H$ is the SM Higgs doublet and $f$ is an energy scale needed to render the coupling constant $c_\psi$ dimensionless. Effective interactions of the form (1) are known to arise in Higgs-portal [28–33] and twin-Higgs models [34–38]. In the former case, the new fermion plays the role of a

dark matter (DM) candidate, while in the latter case, the presence of $\psi$ provides a solution to the hierarchy problem of the Higgs-boson mass in the form of an uncoloured top partner. In fact, in ultraviolet (UV) completions of (1) where the hierarchy problem is addressed, a light Higgs boson is natural if the coupling $c_\psi$ satisfies [39]:

$$|c_\psi| \lesssim \frac{3 y_t^2 f}{2 N_\psi m_\psi} \,. \tag{2}$$

Here $y_t = \sqrt{2} m_t / v \simeq 0.94$ is the top-quark Yukawa coupling with $m_t \simeq 163\,\text{GeV}$ the top-quark $\overline{\text{MS}}$ mass and $v \simeq 246\,\text{GeV}$ is the vacuum expectation value (VEV) of the Higgs field, while $m_\psi$ denotes the mass of the new fermion and we have assumed that $\psi$ transforms under the fundamental representation of a $SU(N_\psi)$ gauge group. Notice that for $m_\psi \simeq y_t f / \sqrt{2}$ and $N_\psi = 3$ which holds in standard twin-Higgs models the naturalness condition (2) simply reads $|c_\psi| \lesssim y_t / \sqrt{2} \simeq 0.7$. This corresponds to a convention in which all Higgs couplings are modified by a factor $1 - v^2 / (2 f^2)$ relative to their SM predictions. Finally, it is important to realise that in twin-Higgs realisations of (1) the UV cut-off $\Lambda$ of the theory is $\Lambda \simeq 4\pi f$, corresponding to the energy scale at which the theory becomes strongly coupled [34–38]. This is the relevant UV cut-off scale in most parts of our work (see Section 3, Section 4, Appendix A, Appendix B and Appendix D), and as it will become clear, all the considered observables are dominated by energies safely below this scale. In contrast, in weakly-coupled Higgs-portal models [28–33] the UV cut-off is $\Lambda \simeq f$, i.e. it is equal to the suppression scale appearing in the Lagrangian (1). Results for the latter case are relegated to Appendix C.

In order to provide a complete picture of the reach that hadron colliders have in the context of the fermionic Higgs portal (1) we consider the high-luminosity option of the LHC (HL-LHC), the high-energy upgrade of the LHC (HE-LHC) and a Future Circular Collider (FCC) with centre-of-mass (CM) energies of 14 TeV, 27 TeV and 100 TeV, respectively. Like in the previous publication [27] that considered the marginal Higgs portal, we focus our attention in the present study on the indirect constraints that measurements of $pp \to h^* \to ZZ$ and $pp \to hh$ production are expected to allow to set. Compared to the marginal Higgs portal we find that in the case of the fermionic Higgs portal the high-energy tails of the relevant kinematic distributions in both off-shell and double-Higgs production are enhanced. This is a result of the higher-dimensional nature of (1), which requires the inclusion of dimension-six operators in the calculation of the $gg \to h^* \to ZZ$ and $gg \to hh$ amplitudes to render them UV finite. The renormalisation group (RG) flow of the corresponding Wilson coefficients in turn leads to logarithmically enhanced corrections that modify the $gg \to h^* \to ZZ$ and $gg \to hh$ matrix elements and therefore the resulting kinematic distributions in a non-trivial fashion.

This logarithmic enhancement makes indirect probes, i.e. processes that test the quantum structure of the theory, in general more powerful than direct tests, i.e. measurements that dominantly test tree-level interactions, when constraining fermionic Higgs-portal interactions of the form (1). In order to emphasise this point we derive the direct constraints that searches for $\psi \bar{\psi}$ production in the VBF Higgs production channel may be able to set at future hadron machines and compare the obtained bounds to the limits that result from the various indirect Higgs probes.

This article is structured as follows: a detailed discussion of all the ingredients necessary to obtain the one-loop corrections to $pp \to h^* \to ZZ$ and $pp \to hh$ production in the context of the fermionic Higgs portal are provided in Section 2. In Section 3 we study the effects that (1) leave in the kinematic distributions of the off-shell and double-Higgs production channel as well as in pair production of portal fermions in the VBF Higgs production channel. The numerical analysis of the HL-LHC, HE-LHC and FCC reach is performed in Section 4 and contains a comparison of the constraints on the model parameter space obtained via various indirect

and direct Higgs probes. We conclude in Section 5. Supplementary material is relegated to a number of appendices.

## 2 Calculation

In this section, we describe the calculation of all the ingredients that are necessary to obtain predictions for the processes $gg \to h^* \to ZZ$ and $gg \to hh$. The actual generation and computation of the relevant amplitudes made use of the Mathematica packages FeynArts [40], FeynRules [41], FormCalc [42, 43] and Package-X [44].

The Lagrangian necessary for the further discussion takes the following form

$$\mathcal{L} = \bar{\psi} \left( i \slashed{D} - m_\psi \right) \psi + \frac{c_\psi}{f} \left( |H|^2 - \frac{v^2}{2} \right) \bar{\psi} \psi + \sum_{i=6,H\square} C_i Q_i \,, \tag{3}$$

where we have assumed that the fermion $\psi$ transforms in the fundamental representation of a dark $SU(N_\psi)$ gauge group with the corresponding covariant derivative $\slashed{D} \equiv D_\mu \gamma^\mu$. Notice that besides the portal coupling (1) the above Lagrangian contains the following two dimension-six operators

$$Q_6 = |H|^6 \,, \qquad Q_{H\square} = \partial_\mu |H|^2 \, \partial^\mu |H|^2 \,. \tag{4}$$

The associated Wilson coefficients $C_6$ and $C_{H\square}$ carry mass dimension $-2$. As we will explain in the next subsection, once radiative corrections on top of (1) are considered, the resulting quantum theory inevitably contains $Q_6$ and $Q_{H\square}$. At the one-loop level and up to dimension six, only the latter two operators are generated, making the Lagrangian (3) the minimal effective field theory (EFT) that allows to consistently calculate off-shell and double-Higgs production in fermionic Higgs-portal models.

### 2.1 RG evolution of the Wilson coefficients

The appearance of the higher-dimensional terms entering (3) is readily understood by noticing that Feynman diagrams of the type shown in Figure 1 lead to UV divergent contributions proportional to the operators $Q_6$ and $Q_{H\square}$. These UV divergences appear as $1/\epsilon$ poles if the respective scattering amplitudes are calculated using dimensional regularisation in $d = 4 - 2\epsilon$ space-time dimensions. They determine the RG evolution of the Wilson coefficients $C_6$ and $C_{H\square}$. At leading-logarithmic order we find in the $\overline{\text{MS}}$ scheme the following result

$$C_i(\mu) = C_i(\mu_*) + \gamma_i \ln\left(\frac{\mu_*^2}{\mu^2}\right) \,. \tag{5}$$

Here $\mu$ is a low-energy scale while $\mu_*$ denotes a high-energy matching scale. The relevant one-loop anomalous dimensions that appear in (5) are given by

$$\gamma_{H\psi,6} = \frac{N_\psi c_\psi^3 m_\psi}{4\pi^2 f^3} \,, \qquad \gamma_{H\psi,H\square} = \frac{N_\psi c_\psi^2}{16\pi^2 f^2} \,. \tag{6}$$

### 2.2 Higgs tadpole

The presence of (3) affects the Higgs potential in such a way that its minimum is shifted. To correct for this shift, one has to perform a renormalisation of the Higgs tadpole

$$\hat{T} = T + \delta t \,, \tag{7}$$

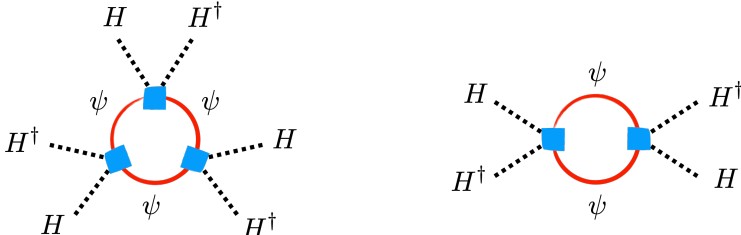

Figure 1: Examplary graphs that lead to UV divergent contributions proportional to the operator $Q_6$ (left) and $Q_{H\square}$ (right), respectively. Insertions of the portal coupling (1) are indicated by the blue squares.

which corresponds to a renormalisation of the Higgs VEV. Here $T$ is the Higgs field one-point amputated Green's function while $\delta t$ is the corresponding counterterm. It originates from the following definition of $t$ in the Higgs potential

$$V \supset -v\left(\mu^2 - v^2\lambda\right)h \equiv -t\,h\,, \tag{8}$$

where $\mu^2$ and $\lambda$ are the Higgs mass parameter and quartic coupling, respectively. At the tree level in the SM, one has $T = t = 0$. In our case, $T$ receives contributions from the left one-loop diagram in Figure 2 as well as a tree-level contribution associated to the operator $Q_6$. We obtain

$$T = \frac{N_\psi c_\psi m_\psi v}{4\pi^2 f}A_0(m_\psi^2) + \frac{3}{4}v^5\left[C_6(\mu) - \frac{\gamma_{H\psi,6}}{\epsilon}\right], \tag{9}$$

where $A_0$ is the one-point Passarino-Veltman (PV) scalar integral defined as in [42,43] and the expression $1/\epsilon$ is an abbreviation for $1/\epsilon - \gamma_E + \ln(4\pi)$ with $\gamma_E \simeq 0.577216$ the Euler constant. Notice that the normalisation of the PV one-loop integrals [42,43] used here contains a factor $(4\pi)^\epsilon e^{-\gamma_E\epsilon} = 1 - \gamma_E + \ln(4\pi)$ which implies that if the $1/\epsilon$ poles cancel so do the additional $\gamma_E$ and $\ln(4\pi)$ terms. This is the reason why in (9) we have not explicitly included these terms. We will do the same hereafter.

The standard renormalisation of the Higgs tadpole consists in defining $\delta t$ such that

$$\hat{T} = 0\,, \tag{10}$$

order by order in perturbation theory. This choice minimises the effective potential of the Higgs field and in our case leads to the following expression

$$\delta t = -\frac{N_\psi c_\psi m_\psi^3 v}{4\pi^2 f}\left[\frac{1}{\epsilon} + 1 + \ln\left(\frac{\mu^2}{m_\psi^2}\right)\right] - \frac{3}{4}v^5\left\{C_6(\mu_*) - \frac{N_\psi c_\psi^3 m_\psi}{4\pi^2 f^3}\left[\frac{1}{\epsilon} - \ln\left(\frac{\mu_*^2}{\mu^2}\right)\right]\right\}\,, \tag{11}$$

for the tadpole counterterm.

## 2.3 Higgs-boson self-energy

In the theory described by (3), the renormalised self-energy of the Higgs boson takes the following form

$$\hat{\Sigma}(\hat{s}) = \Sigma(\hat{s}) + \left(\hat{s} - m_h^2\right)\delta Z_h - \delta m_h^2 + \frac{15}{4}v^4\left[C_6(\mu) - \frac{\gamma_{H\psi,6}}{\epsilon}\right] + 2v^2\hat{s}\left[C_{H\square}(\mu) - \frac{\gamma_{H\psi,H\square}}{\epsilon}\right], \tag{12}$$

where $\hat{s} = p^2$ with $p$ the external four-momentum entering the Higgs-boson propagator and $m_h \simeq 125\,\text{GeV}$ is the Higgs mass. Furthermore, $\Sigma(\hat{s})$ denotes the bare one-loop Higgs-boson

self-energy, $\delta Z_h$ and $\delta m_h^2$ represent the one-loop corrections to the Higgs wave function and mass counterterm and the terms in the second line are the tree-level $\overline{\text{MS}}$ counterterm contributions associated with the operators $Q_6$ and $Q_{H\square}$. The results for the Wilson coefficients and the anomalous dimensions can be found in (5) and (6), respectively. Notice that in (12) we have not included a Higgs tadpole contribution because it is exactly cancelled by the respective counterterm for the choice (10) of tadpole renormalisation.

At the one-loop level the bare Higgs-boson self-energy receives contributions from Feynman graphs such as the ones depicted on the right-hand side in Figure 2. We find

$$\Sigma(\hat{s}) = \frac{N_\psi c_\psi}{4\pi^2 f}\left(m_\psi - \frac{c_\psi v^2}{f}\right)A_0(m_\psi^2) + \frac{N_\psi c_\psi^2 v^2}{8\pi^2 f^2}\left(\hat{s} - 4m_\psi^2\right)B_0\left(\hat{s}, m_\psi^2, m_\psi^2\right), \quad (13)$$

where $B_0$ is the two-point PV scalar integral defined as in [42, 43].

The wave function renormalisation (WFR) constant $\delta Z_h$ and the mass counterterm $\delta m_h^2$ are fixed by imposing the on-shell renormalisation conditions

$$\hat{\Sigma}(m_h^2) = 0, \qquad \hat{\Sigma}'(m_h^2) = \frac{d\hat{\Sigma}(\hat{s})}{d\hat{s}}\bigg|_{\hat{s}=m_h^2} = 0. \quad (14)$$

Using (12) and (13) as well as requiring (14), we obtain the following expressions

$$\delta Z_h = -\frac{N_\psi c_\psi^2 v^2}{8\pi^2 f^2}\left\{\left[B_0(m_h^2, m_\psi^2, m_\psi^2) - \frac{1}{\epsilon} + \ln\left(\frac{\mu_*^2}{\mu^2}\right)\right]\right.$$

$$\left. + \left(m_h^2 - 4m_\psi^2\right)B_0'(m_h^2, m_\psi^2, m_\psi^2)\right\} - 2v^2 C_{H\square}(\mu_*),$$

$$\delta m_h^2 = \frac{N_\psi c_\psi}{4\pi^2 f}\left(m_\psi - \frac{c_\psi v^2}{f}\right)A_0(m_\psi^2) + \frac{N_\psi c_\psi^2 v^2}{8\pi^2 f^2}\left(m_h^2 - 4m_\psi^2\right)B_0(m_h^2, m_\psi^2, m_\psi^2) \quad (15)$$

$$+ \frac{15 v^4}{4}\left\{C_6(\mu_*) - \frac{N_\psi c_\psi^3 m_\psi}{4\pi^2 f^3}\left[\frac{1}{\epsilon} - \ln\left(\frac{\mu_*^2}{\mu^2}\right)\right]\right\}$$

$$+ 2v^2 m_h^2\left\{C_{H\square}(\mu_*) - \frac{N_\psi c_\psi^2}{16\pi^2 f^2}\left[\frac{1}{\epsilon} - \ln\left(\frac{\mu_*^2}{\mu^2}\right)\right]\right\},$$

where $B_0'$ denotes the derivative of the two-point PV scalar integral with respect to the kinematic invariant as defined in (14). Notice that the sum of the three terms that appear in the square bracket of $\delta Z_h$ are UV finite, because the explicit $1/\epsilon$ pole cancels against the UV divergence of the two-point PV scalar integral $B_0$. Explicitly, we find

$$\delta Z_h = -\frac{N_\psi c_\psi^2 v^2}{8\pi^2 f^2}\left[\left(1 + \frac{2m_\psi^2}{m_h^2}\right)\Lambda(m_h^2, m_\psi, m_\psi) + 1 + \frac{4m_\psi^2}{m_h^2} + \ln\left(\frac{\mu_*^2}{m_\psi^2}\right)\right]$$

$$- 2v^2 C_{H\square}(\mu_*), \quad (16)$$

where $\Lambda(\hat{s}, m_0, m_1)$ represents the part of the $B_0(\hat{s}, m_0^2, m_1^2)$ integral containing the $\hat{s}$-plane branch cut

$$\Lambda(\hat{s}, m_0, m_1) = \frac{\sqrt{\lambda(m_0^2, m_1^2, \hat{s})}}{\hat{s}}\ln\left(\frac{m_0^2 + m_1^2 - \hat{s} + \sqrt{\lambda(m_0^2, m_1^2, \hat{s})}}{2m_0 m_1}\right), \quad (17)$$

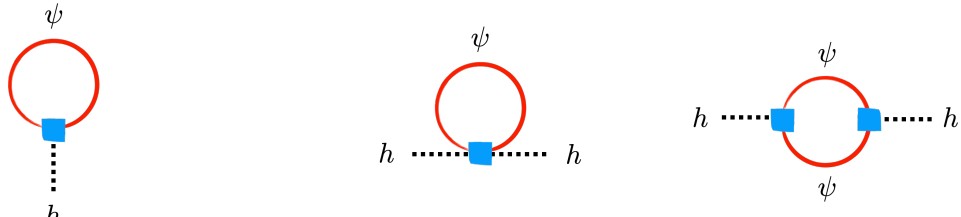

Figure 2: One-loop contributions to the Higgs tadpole (left) and Higgs-boson self-energy (right). The blue squares indicate insertions of the portal coupling (1).

with

$$\lambda(a,b,c) = a^2 - 2a(b+c) + (b-c)^2 \,, \tag{18}$$

the Källén kinematic polynomial.

The results (13) and (15) can be combined to obtain the renormalised self-energy of the Higgs boson (12). In terms of (17) we arrive at

$$\hat{\Sigma}(\hat{s}) = \frac{N_\psi c_\psi^2 v^2}{8\pi^2 f^2} \left\{ \left(\hat{s} - 4m_\psi^2\right)\left[\Lambda(\hat{s}, m_\psi, m_\psi) - \Lambda(m_h^2, m_\psi, m_\psi)\right] \right.$$
$$\left. - \left(\hat{s} - m_h^2\right)\left[\frac{2m_\psi^2}{m_h^2}\Lambda(m_h^2, m_\psi, m_\psi) - 1 + \frac{4m_\psi^2}{m_h^2}\right]\right\}. \tag{19}$$

Notice that our result for the renormalised Higgs-boson self-energy is both $\mu_*$ and $\mu$ independent. In particular, the expression (19) does neither depend on the initial conditions $C_6(\mu_*)$ and $C_{H\square}(\mu_*)$ of the Wilson coefficients nor on the logarithm $\ln\left(\mu_*^2/\mu^2\right)$.

## 2.4 Off-shell Higgs production

At the one-loop level the $gg \to h^* \to ZZ$ process receives contributions from Feynman graphs such as the one displayed in Figure 3 that contains a modified Higgs propagator with two insertions of the portal coupling (1). The full BSM amplitude for off-shell Higgs production in the $gg \to h^* \to ZZ$ channel can be factorised as follows

$$\mathcal{A}(gg \to h^* \to ZZ) = \left[1 + \sigma(\hat{s})\right]\mathcal{A}_{\mathrm{SM}}(gg \to h^* \to ZZ)\,, \tag{20}$$

where $\mathcal{A}_{\mathrm{SM}}(gg \to h^* \to ZZ)$ is the corresponding SM amplitude. The $\hat{s}$-dependent form factor appearing in (20) receives a contribution from the Higgs WFR constant $\delta Z_h$ and the renormalised self-energy of the Higgs boson $\hat{\Sigma}(\hat{s})$. Explicitly, one has

$$\sigma(\hat{s}) = \delta Z_h - \frac{\hat{\Sigma}(\hat{s})}{\hat{s} - m_h^2}$$

$$= -\frac{N_\psi c_\psi^2 v^2}{8\pi^2 f^2}\left[\frac{\left(\hat{s} - 4m_\psi^2\right)\Lambda(\hat{s}, m_\psi, m_\psi) - \left(m_h^2 - 4m_\psi^2\right)\Lambda(m_h^2, m_\psi, m_\psi)}{\hat{s} - m_h^2}\right.$$

$$\left. + 2 + \ln\left(\frac{\mu_*^2}{m_\psi^2}\right)\right] - 2v^2 C_{H\square}(\mu_*), \tag{21}$$

where in order to arrive at the final result we have used (16) and (19). Notice that the explicit contribution of the Higgs WFR constant $\delta Z_h$ coming from the vertices exactly cancels against

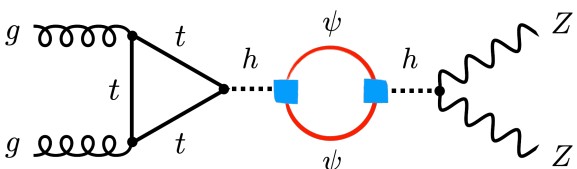

Figure 3: One-loop contribution to off-shell Higgs production in the $gg \to h^* \to ZZ$ channel. Insertions of the portal coupling (1) are indicated by the blue squares.

the $\delta Z_h$ piece present in the renormalised self-energy of the Higgs boson $\hat{\Sigma}(\hat{s})$. In contrast, the Higgs WFR constant $\delta Z_h$ does not drop out in the on-shell Higgs signal strengths to be discussed later.

It is important to realise that in the limit $\hat{s} \gg m_h^2, m_\psi^2$ the $\hat{s}$-dependent form factor (21) behaves as

$$\sigma(\hat{s}) \simeq -\frac{N_\psi c_\psi^2 v^2}{8\pi^2 f^2} \left[ 2 + \ln\left(\frac{\mu_*^2}{\hat{s}}\right) + i\pi \right] - 2v^2 C_{H\square}(\mu_*), \qquad (22)$$

which depends logarithmically on the high-energy matching scale $\mu_*$ and linearly on the initial condition $C_{H\square}(\mu_*)$. This is a consequence of the RG flow of the operator $Q_{H\square}$ discussed in Section 2.1. As we will see below the logarithmic enhancement of $\sigma(\hat{s})$ and therefore $\mathcal{A}(gg \to h^* \to ZZ)$ will play a crucial role in our numerical analysis of the constraints on (1) that future measurements of off-shell Higgs production are expected to be able to set.

## 2.5 Double-Higgs production

In Figure 4 two example graphs are shown that give rise to double-Higgs production via $gg \to hh$ in the presence of the portal coupling (1). The corresponding complete amplitude for double-Higgs production can be written as

$$\mathcal{A}(gg \to hh) = \left[1 + \delta Z_h\right] \mathcal{A}_{\text{SM}}^{\square}(gg \to hh) + \left[1 + \delta(\hat{s})\right] \mathcal{A}_{\text{SM}}^{\triangle}(gg \to hh), \qquad (23)$$

where $\mathcal{A}_{\text{SM}}^{\square}(gg \to hh)$ and $\mathcal{A}_{\text{SM}}^{\triangle}(gg \to hh)$ denote the $gg \to hh$ amplitude arising from box and triangle diagrams in the SM, respectively. The $\hat{s}$-dependent form factor $\delta(\hat{s})$ encodes the one-loop and tree-level corrections associated with (3) and effectively corresponds to a modification of the trilinear Higgs self coupling. The box contribution to the amplitude instead only receives a correction from the Higgs wave function due to the two final-state Higgs bosons being on-shell.

The first ingredient needed to determine $\delta(\hat{s})$ is the renormalised trilinear Higgs vertex. It takes the following form

$$\hat{\Gamma}(\hat{s}) = \Gamma(\hat{s}) - \frac{3}{v}\left[\frac{\delta t}{v} + \delta m_h^2 + \frac{3}{2}m_h^2 \delta Z_h\right] + 15\,v^3\left[C_6(\mu) - \frac{\gamma_{H\psi,6}}{\epsilon}\right]$$
$$+ 2v\left(\hat{s} + 2m_h^2\right)\left[C_{H\square}(\mu) - \frac{\gamma_{H\psi,H\square}}{\epsilon}\right], \qquad (24)$$

where $\Gamma(\hat{s})$ denotes the bare trilinear Higgs vertex, $\delta t$ is the one-loop Higgs tadpole counterterm, $\delta Z_h$ and $\delta m_h^2$ encode the one-loop corrections to the Higgs wave function and the mass counterterm, respectively, and the contributions in the second line are the one-loop $\overline{\text{MS}}$ counterterm corrections that arise from $Q_6$ and $Q_{H\square}$. Note that the contribution from the tadpole counterterm can be found using its definition (8). The requirement $\hat{T} = 0$ preserves the tree-level relations between the Higgs VEV and the parameters in the Higgs potential, such that one finds that the counterterm for the Higgs quartic can be expressed as

$\delta\lambda = \delta m_h^2/(2v^2) + \delta t/(2v^3)$. The bare trilinear Higgs vertex entering (24) is given by

$$\Gamma(\hat{s}) = -\frac{N_\psi c_\psi^2 v}{8\pi^2 f^2}\Big[ 6A_0(m_\psi^2) - 2\left(m_h^2 - 4m_\psi^2\right)B_0\left(m_h^2, m_\psi^2, m_\psi^2\right)$$
$$-\left(\hat{s} - 4m_\psi^2\right)B_0\left(\hat{s}, m_\psi^2, m_\psi^2\right)\Big]$$
$$+\frac{N_\psi c_\psi^3 m_\psi v^3}{4\pi^2 f^3}\Big[ 4B_0\left(m_h^2, m_\psi^2, m_\psi^2\right) + 2B_0\left(\hat{s}, m_\psi^2, m_\psi^2\right)$$
$$-\left(\hat{s} + 2m_h^2 - 8m_\psi^2\right)C_0\left(\hat{s}, m_h^2, m_h^2, m_\psi^2, m_\psi^2, m_\psi^2\right)\Big],$$
(25)

where $C_0$ denotes the three-point PV scalar integral defined as in [42, 43]. Inserting the expressions (5), (11), (15) and (25) into (24) leads to

$$\hat{\Gamma}(\hat{s}) = \frac{N_\psi c_\psi^2 v}{16\pi^2 f^2}\Bigg[ \left(7m_h^2 + 26m_\psi^2\right)\Lambda(m_h^2, m_\psi, m_\psi) + 2\left(\hat{s} - 4m_\psi^2\right)\Lambda(\hat{s}, m_\psi, m_\psi)$$
$$+ 4\hat{s} + 5m_h^2 + 36m_\psi^2 + \left(2\hat{s} + 7m_h^2\right)\ln\left(\frac{\mu_*^2}{m_\psi^2}\right)\Bigg]$$
$$+\frac{N_\psi c_\psi^3 m_\psi v^3}{4\pi^2 f^3}\Bigg[ 4\Lambda(m_h^2, m_\psi, m_\psi) + 2\Lambda(\hat{s}, m_\psi, m_\psi)$$
$$-\left(\hat{s} + 2m_h^2 - 8m_\psi^2\right)\Omega(\hat{s}, m_h, m_\psi) + 12 + 6\ln\left(\frac{\mu_*^2}{m_\psi^2}\right)\Bigg]$$
$$+ 6v^3 C_6(\mu_*) + v\left(2\hat{s} + 7m_h^2\right)C_{H\Box}(\mu_*),$$
(26)

where

$$\Omega(\hat{s}, m_h, m_\psi) = \lim_{\varepsilon \to 0^+}\int_0^1 dx \int_0^{1-x} dy \left[\hat{s}x(1-x-y) + m_h^2 y(1-y) - m_\psi^2 + i\varepsilon\right]^{-1}. \quad (27)$$

Notice that our result (26) for the renormalised trilinear Higgs vertex is UV finite but depends on the high-energy matching scale $\mu_*$ both explicitly and through the initial conditions $C_6(\mu_*)$ and $C_{H\Box}(\mu_*)$. This is again a consequence of the RG evolution of the operators $Q_6$ and $Q_{H\Box}$ that has been discussed in Section 2.1.

The $\hat{s}$-dependent form factor introduced in (24) receives contributions from the Higgs WFR constant $\delta Z_h$, the renormalised self-energy of the Higgs boson $\hat{\Sigma}(\hat{s})$ and the renormalised tri-

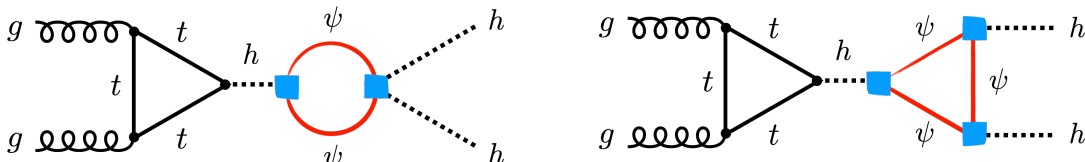

Figure 4: Two example one-loop diagrams that lead to double-Higgs production via $gg \to hh$. Each blue square represents an insertions of the portal coupling (1).

linear Higgs vertex $\hat{\Gamma}(\hat{s})$. To leading order we find the following expression

$$
\begin{aligned}
\delta(\hat{s}) = {} & \frac{1}{2}\delta Z_h - \frac{\hat{\Sigma}(\hat{s})}{\hat{s} - m_h^2} - \frac{\hat{\Gamma}(\hat{s})}{6v\lambda_{\mathrm{SM}}} \\
= {} & \frac{N_\psi c_\psi^2 v^2}{16\pi^2 f^2}\left[\frac{10m_h^4 - 4m_h^2(\hat{s} + m_\psi^2) - 20m_\psi^2\hat{s}}{3m_h^2(\hat{s} - m_h^2)}\Lambda(m_h^2, m_\psi, m_\psi)\right. \\
& - \frac{2(\hat{s} + 2m_h^2)(\hat{s} - 4m_\psi^2)}{3m_h^2(\hat{s} - m_h^2)}\Lambda(\hat{s}, m_\psi, m_\psi) \\
& \left. - \frac{14}{3} - \frac{4(\hat{s} + 6m_\psi^2)}{3m_h^2} - \left(\frac{10}{3} + \frac{2\hat{s}}{3m_h^2}\right)\ln\left(\frac{\mu_*^2}{m_\psi^2}\right)\right] \\
& - \frac{N_\psi c_\psi^3 m_\psi v^4}{4\pi^2 m_h^2 f^3}\left[\frac{4}{3}\Lambda(m_h^2, m_\psi, m_\psi) + \frac{2}{3}\Lambda(\hat{s}, m_\psi, m_\psi)\right. \\
& \left. - \frac{1}{3}(\hat{s} + 2m_h^2 - 8m_\psi^2)\Omega(\hat{s}, m_h, m_\psi) + 4 + 2\ln\left(\frac{\mu_*^2}{m_\psi^2}\right)\right] \\
& - \frac{2v^4}{m_h^2}C_6(\mu_*) - \frac{2v^2}{3m_h^2}(\hat{s} + 5m_h^2)C_{H\square}(\mu_*).
\end{aligned}
\tag{28}
$$

To arrive at the final result we have used that in the SM the Higgs trilinear self coupling is given by $\lambda_{\mathrm{SM}} = m_h^2/(2v^2)$ and employed the expressions (16), (19) and (26).

It is again instructive to consider the limit $\hat{s} \gg m_h^2, m_\psi^2$ of the $\hat{s}$-dependent form factor (28). We obtain

$$
\delta(\hat{s}) \simeq -\frac{N_\psi c_\psi^2 v^2 \hat{s}}{24\pi^2 m_h^2 f^2}\left[2 + \ln\left(\frac{\mu_*^2}{\hat{s}}\right) + i\pi\right] - \frac{2v^2\hat{s}}{3m_h^2}C_{H\square}(\mu_*).
\tag{29}
$$

Notice that this result grows linearly with $\hat{s}$ and like (22) depends logarithmically on the high-energy matching scale $\mu_*$ and linearly on the initial condition $C_{H\square}(\mu_*)$. In general terms the result (29) implies that the one-loop $gg \to hh$ amplitude in the fermionic Higgs-portal model (3) violates perturbative unitarity for sufficiently large $\hat{s}$. To mitigate this issue, we use in this article only the total $pp \to hh$ production cross section, which is dominated by the contributions close to the double-Higgs threshold $\hat{s} = 4m_h^2$, when constraining (1).

## 3 Numerical analysis

In this section, we discuss the kinematic distributions of off-shell, double- and VBF Higgs production that we will use to study the sensitivity of future hadron colliders to the fermionic

Higgs portal. We perform our off-shell and double-Higgs analyses along the lines of the article [27], while in the case of pair production of the new fermions in the VBF Higgs production channel we rely on the publication [15].

## 3.1 Off-shell Higgs production

In order to compute kinematic distributions for $pp \to ZZ \to 4\ell$ production, we incorporated the formulas (20) and (21) into the event generator MCFM 8.0 [45]. In addition to the four-lepton invariant mass ($m_{4\ell}$) spectrum, our MCFM implementation can evaluate the following matrix-element (ME) based kinematic discriminant

$$D_S = \log_{10}\left(\frac{P_h}{P_{gg} + c\,P_{q\bar{q}}}\right),\tag{30}$$

which has also been employed for example in the publications [46–48]. Here, $P_h$ represents the squared ME for the $gg \to h^* \to ZZ \to 4\ell$ process, while $P_{gg}$ is the squared ME encompassing all $gg$-initiated channels, including the Higgs channel, the continuum background and their interference. Additionally, $P_{q\bar{q}}$ denotes the squared ME for the $q\bar{q} \to ZZ \to 4\ell$ process. Like in [46–48], the constant $c$ in (30) is chosen as 0.1 to balance the contributions from $q\bar{q}$- and $gg$-initiated processes. In the SM, over 99% of the total $pp \to ZZ \to 4\ell$ cross section is observed within the range of $-4.5 < D_S < 0.5$ [46]. Consequently, for BSM scenarios predicting events with $D_S < -4.5$ or $D_S > 0.5$, the variable $D_S$ serves as a null test. See for example [49,50] for BSM search strategies that exploit this feature.

Following the methodology outlined in [27,49], we incorporate QCD corrections into our $pp \to ZZ \to 4\ell$ analysis. Specifically, for the two distinct production channels, we compute the so-called $K$-factor, defined as the ratio between the fiducial cross section at a given order in QCD and the corresponding leading order (LO) prediction. For the $gg$-initiated contribution, we rely on the results from [51–54]. The ratio between the next-to-leading (NLO) and LO gluon-gluon-fusion predictions remains essentially constant across $m_{4\ell}$, and through averaging, we determine $K_{gg}^{\mathrm{NLO}} = 1.83$. In the case of the $q\bar{q}$-initiated contribution, we utilise the next-to-next-to-leading order (NNLO) results obtained in [52,55]. The relevant $K$-factor is also observed to be nearly flat in $m_{4\ell}$, with a central value of $K_{q\bar{q}}^{\mathrm{NNLO}} = 1.55$. These $K$-factors are then employed to derive a QCD-improved prediction for the $pp \to ZZ \to 4\ell$ cross section differentially in the variable $O$ by means of:

$$\frac{d\sigma_{pp}}{dO} = K_{gg}^{\mathrm{NLO}}\left(\frac{d\sigma_{gg}}{dO}\right)_{\mathrm{LO}} + K_{q\bar{q}}^{\mathrm{NNLO}}\left(\frac{d\sigma_{q\bar{q}}}{dO}\right)_{\mathrm{LO}}.\tag{31}$$

It is important to emphasise that (31) strictly holds only for the $m_{4\ell}$ spectrum. As shown in [49], when applying (31) to the $D_S$ distribution one obtains a nearly constant $K$-factor of approximately 1.6 between the LO and the improved result. The inclusion of higher-order QCD corrections furthermore reduces the scale uncertainties of $D_S$ by a factor of about 3 from around 7.5% to 2.5%. The quoted uncertainties rely on (31) in the case of the QCD-improved prediction of $D_S$ and have been obtained from seven-point scale variations enforcing the constraint $1/2 \leq \mu_R/\mu_F \leq 2$ on the renormalisation and factorisation scales $\mu_R$ and $\mu_F$. To which extent the small scale uncertainties of the QCD-improved prediction (31) provide a reliable estimate of the size of higher-order QCD effects in $D_S$ is questionable. As a result, in our exploration of the collider reach in Section 4, we will adopt different assumptions regarding the systematic uncertainties involved in our ME-based search strategy. It is worth noting that a similar approach is employed in the projections [56,57] that assess the high-luminosity LHC (HL-LHC) potential for constraining off-shell Higgs-boson production and the Higgs-boson total width in $pp \to ZZ \to 4\ell$.

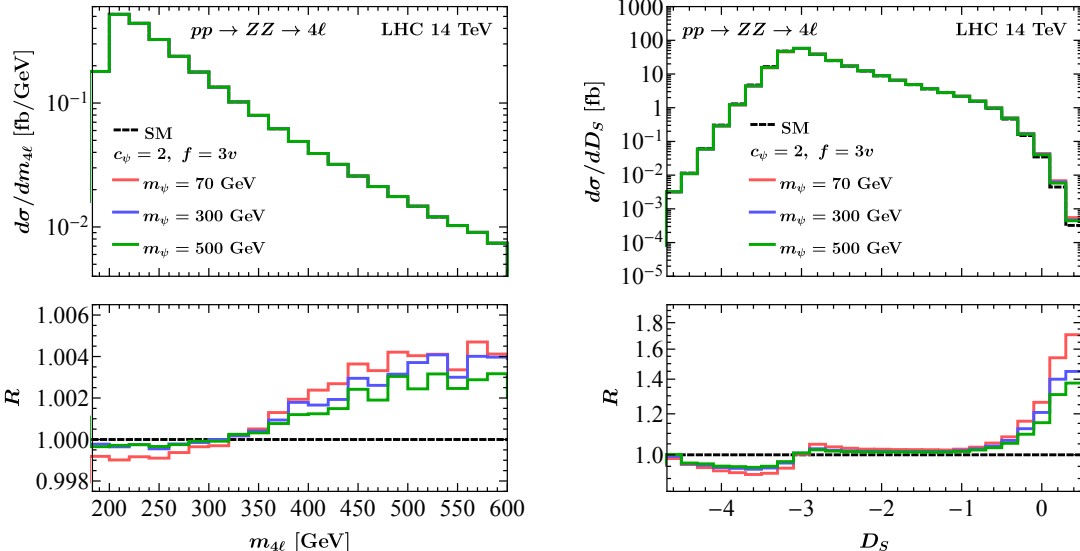

Figure 5: Comparison of off-shell Higgs production distributions for the SM (dashed black) and three distinct fermion Higgs-portal scenarios (1). These scenarios assume $c_\psi = 2$ and $f = 3v$, with fermion masses set to $m_\psi = 70\,\text{GeV}$ (solid red), $m_\psi = 300\,\text{GeV}$ (solid blue) and $m_\psi = 500\,\text{GeV}$ (solid green). The left (right) plot presents results for the four-lepton invariant mass ($m_{4\ell}$) and the discriminant variable ($D_S$), respectively, in $pp \to ZZ \to 4\ell$ production. All distributions are based on QCD-improved predictions and pertain to LHC collisions at a CM energy of $\sqrt{s} = 14\,\text{TeV}$. The lower panels illustrate the ratios between the BSM distributions and their corresponding SM predictions. Further details are provided in the main text.

In our HL-LHC analysis of $pp \to ZZ \to 4\ell$ production, we focus on four-lepton invariant masses within the range of $140\,\text{GeV} < m_{4\ell} < 600\,\text{GeV}$. The charged leptons are required to satisfy the requirement $|\eta_\ell| < 2.5$ on their pseudorapidity. Additionally, the lepton with the highest transverse momentum ($p_T$) must obey $p_{T,\ell_1} > 20\,\text{GeV}$, while the second, third and fourth hardest leptons are required to meet the conditions $p_{T,\ell_2} > 15\,\text{GeV}$, $p_{T,\ell_3} > 10\,\text{GeV}$ and $p_{T,\ell_4} > 6\,\text{GeV}$, respectively. The lepton pair with an invariant mass closest to the $Z$-boson mass is constrained to lie in the range $50\,\text{GeV} < m_{12} < 106\,\text{GeV}$, while the remaining lepton pair must have an invariant mass within $50\,\text{GeV} < m_{34} < 115\,\text{GeV}$. We note that similar cuts have been employed in the ATLAS and CMS analyses [46–48,56–59]. For the selected leptons, we assume a detection efficiency of 99% (95%) for muons (electrons). These efficiencies correspond to those reported in the latest ATLAS analysis of off-shell Higgs production [48]. As input we use $G_F = 1/(\sqrt{2}v^2) = 1.16639 \cdot 10^{-5}\,\text{GeV}^{-2}$, $m_Z = 91.1876\,\text{GeV}$, $m_h = 125\,\text{GeV}$ and $m_t = 173\,\text{GeV}$. The `NNPDF40_nlo_as_01180` parton distribution functions (PDFs) [60] are employed. The renormalisation and factorisation scales are dynamically set to $m_{4\ell}$ on an event-by-event basis. Our $pp \to ZZ \to 4\ell$ predictions encompass both different-flavour ($e^+e^-\mu^+\mu^-$) and same-flavour ($2e^+2e^-$ and $2\mu^+2\mu^-$) decay channels.

In Figure 5, we compare the $m_{4\ell}$ (left) and $D_S$ (right) distributions in $pp \to ZZ \to 4\ell$ production at $\sqrt{s} = 14\,\text{TeV}$. The SM (dashed black) is contrasted with three distinct fermion Higgs-portal scenarios (1) featuring $c_\psi = 2$ and $f = 3v$, alongside the choices $m_\psi = 70\,\text{GeV}$ (solid red), $m_\psi = 300\,\text{GeV}$ (solid blue) and $m_\psi = 500\,\text{GeV}$ (solid green). The plots are generated using $N_\psi = 3$, $\mu_* = 4\pi f$ and $C_{H\Box}(\mu_*) = 0$ in (21). We note that $N_\psi = 3$ is characteristic of

standard twin-Higgs models, while the other two choices imply the absence of a direct matching correction to $Q_{H\square}$ at the UV cut-off scale $\Lambda = 4\pi f$, where the theory becomes strongly coupled. The displayed results therefore exclusively capture the model-independent logarithmic corrections associated with the RG evolution of the Wilson coefficient $C_{H\square}$ from $\mu_*$ down to $m_\psi$ $\left(\text{cf. (5) and (6)}\right)$. Notice that these logarithmically enhanced effects represent the minimal contributions to any UV-finite $gg \to h^* \to ZZ$ amplitude of the form (20). The non-logarithmic contributions associated to $C_{H\square}(\mu_*)$ are instead not determined by the UV-pole structure of $\mathcal{A}(gg \to h^* \to ZZ)$ which renders them model-dependent. By comparing the relative modifications in the panels of Figure 5, it becomes evident that the four-lepton invariant mass $m_{4\ell}$ has a much weaker discriminatory power compared to the variable $D_S$ in constraining interactions of the form (1). This observation aligns with similar findings in the article [27], which investigated the marginal Higgs portal. However, for the fermionic Higgs portal, the largest relative modifications occur in the tails of the distributions for $m_{4\ell} \gtrsim 400\,\text{GeV}$ and $D_S \gtrsim -1$, resulting from the non-decoupling behaviour of (22). Notice also that for the parameters chosen in Figure 5, the upper limit of the probed four-lepton invariant masses $m_{4\ell} < 600\,\text{GeV}$ is safely below the UV cut-off $\Lambda = 4\pi f \simeq 9\,\text{TeV}$ of the twin-Higgs realisations of (1). Under the assumption that the UV completion that leads to the fermionic Higgs portal is strongly-coupled at the scale $\Lambda \simeq 4\pi f$, the tail of the $m_{4\ell}$ spectrum can therefore be reliably computed using the effective Lagrangian (3). Lastly, it is worth noting that the shapes of the $m_{4\ell}$ and $D_S$ distributions obtained in the fermionic Higgs-portal model resemble, to a first approximation, the corresponding spectra in models where the total Higgs width $\Gamma_h$ is modified but not the on-shell Higgs signal strengths — see Figure 8 in Appendix A of the publication [49].

## 3.2 Double-Higgs production

In order to be able to calculate cross sections for double-Higgs production the analytic results (16), (23) and (26) are implemented into MCFM 8.0. The relevant SM amplitudes are thereby taken from [61]. The graph depicted in Figure 6 illustrates how the signal strength ($\mu_{hh}$) in the $pp \to hh$ channel varies with the Wilson coefficient $c_\psi$ assuming a CM energy of $\sqrt{s} = 14\,\text{TeV}$. The used SM parameters and PDFs are identical to those employed in Section 3.1. However, in contrast to adjusting the renormalisation and factorisation scales individually for each event, we have set both scales to a constant value of $2m_h$. The presented fermion Higgs-portal models in (1) are based on the assumptions $f = 3v$ and $m_\psi$ taking on values of $70\,\text{GeV}$ (dashed red), $300\,\text{GeV}$ (dashed blue) and $500\,\text{GeV}$ (dashed green). The parameters $N_\psi = 3$, $\mu_* = 4\pi f$, $C_6(\mu_*) = C_{H\square}(\mu_*) = 0$ in (16) and (26) were utilised to derive the plotted $\mu_{hh}$ values. It is important to stress again that the choice $C_6(\mu_*) = C_{H\square}(\mu_*) = 0$ guarantees that the presented results remain unaffected by the particular UV completion of (1). This means that the results depend only on the logarithmically enhanced RG contributions, which are precisely calculable within the low-energy theory (3). In fact, the contributions to (16) and (26) solely arise from short-distance physics associated with scales between $\mu_* = 4\pi f$ and $m_\psi$. Notice finally that the total $pp \to hh$ cross section is dominated by the contributions close to the double-Higgs threshold $2m_h \simeq 250\,\text{GeV}$. Assuming that the UV completion that leads to (1) becomes strongly-coupled at $\Lambda \simeq 4\pi f \simeq 9\,\text{TeV}$, the signal strength $\mu_{hh}$ in double-Higgs production can hence be calculated with confidence using the effective Lagrangian (3).

The orange regions are excluded by the CMS projection on the signal strength in double-Higgs production at the HL-LHC [62], implying $\mu_{hh} \in [0.7, 1.8]$ at the 95% confidence level (CL). Two notable features of the depicted $pp \to hh$ predictions warrant discussion. First, due to the $c_\psi^3$ and $c_\psi^2$ dependence of (26), the signal strengths are not symmetric under $c_\psi \leftrightarrow -c_\psi$. Second, while the functional form of the signal strength in double-Higgs production depends on the precise value of the mass $m_\psi$, one can observe that in all depicted cases, $\mu_{hh}$ exhibits a pronounced minimum at negative $c_\psi$ values and a shallow minimum at $c_\psi = 0$.

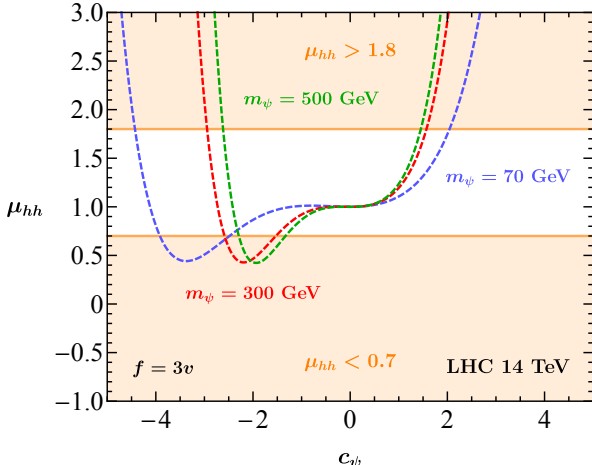

Figure 6: Signal strength for double-Higgs production at $\sqrt{s} = 14\,\text{TeV}$ as a function of the Wilson coefficient $c_\psi$ in (1). The various coloured curves correspond to fermionic Higgs-portal scenarios with $f = 3v$ and the three different choices $m_\psi = 70\,\text{GeV}$ (dashed red), $m_\psi = 300\,\text{GeV}$ (dashed blue) and $m_\psi = 500\,\text{GeV}$ (dashed green). The orange-shaded regions represent areas excluded by the hypothetical experimental 95% CL constraint $\mu_{hh} \in [0.7, 1.8]$. Further details are provided in the main text.

It is important to realise that if the value of $\mu_{hh}$ at its minimum with $c_\psi < 0$ is incompatible with the experimentally allowed range, as is the case for all three $m_\psi$ values shown in Figure 6, adjusting the value of $c_\psi$ will always yield $\mu_{hh}$ values consistent with the experimental data. As we will see in Section 4, the functional dependence of $\mu_{hh}$ on both $c_\psi$ and $m_\psi$ leads to non-trivial shapes of the constraints on the fermionic Higgs portal (1) from future hadron collider measurements of $pp \to hh$ production. In particular, we find that for $c_\psi < 0$, there always exists a funnel in the $m_\psi$–$|c_\psi|$ plane which cannot be excluded by double-Higgs production because the signal strength $\mu_{hh}$ is SM-like.

## 3.3 VBF Higgs production

Under the assumption that the additional fermions are stable on collider timescales or that they decay further into invisible particles, $\psi$ pair production can be tested by looking for missing transverse energy ($E_{T,\text{miss}}$) signatures. In our work we focus on the VBF Higgs production channel since it was found to be more sensitive at hadron colliders than the mono-jet and $t\bar{t}h$ production modes for the marginal and derivative Higgs portals [12, 15, 19]. An example Feynman graph leading to the pair production of the new fermions in the VBF Higgs production channel is depicted in Figure 7.

We have generated both the signal and all the SM background processes at the parton level with `MadGraph5_aMCNLO` [63], showered the obtained events with `Pythia 8.2` [64] and passed them through the `Delphes 3` [65] fast detector simulation. The fermionic Higgs-portal predictions are generated at LO in QCD. The main background channels are $Z$ + jets and $W$ + jets production followed by the $Z \to \nu\bar{\nu}$ and $W \to \ell\nu$ decays, respectively. In both cases the jets can arise from QCD, like in the case of Drell-Yan (DY) production, or through electroweak (EW) interactions, like in the case of VBF gauge-boson production. For strong production we have generated $V$ + 2 jets samples at NLO, while in the EW case our analysis relies on MLM matched [66] LO samples of $V$ + 2 jets and $V$ + 3 jets production. The interfer-

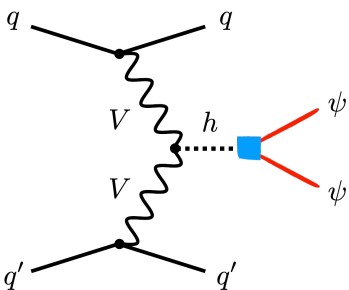

Figure 7: An example diagram leading to the pair production of the new fermions in the VBF Higgs production channel. The portal coupling (1) is indicated by the blue square.

ence between QCD and EW $V + $ jets production turns out to be tiny [67] and thus we have neglected it. Another relevant background is due to $t\bar{t}$ production. We have generated this background at LO including up to two jets using MLM matching. The normalisation of the $V + $ jets backgrounds is fixed by applying a rescaling factor that reproduces event yields of the CMS shape analysis [68] and the $t\bar{t}$ background is normalised to the state-of-the-art SM cross section computations [69,70] that include NNLO and next-to-next-to-leading logarithmic effects. All LO and NLO samples have been generated with `NNPDF2.3LO` and `NNPDF2.3NLO` PDFs [71], respectively. For more information about the background simulation and validation see [15].

In our physics analysis we apply the following set of baseline cuts to both the signal and the background events:

$$
E_{T,\text{miss}} > 80\,\text{GeV}, \quad N_j \geq 2, \quad p_{T,j_{1,2}} > 50\,\text{GeV}, \quad \left|\eta_{j_{1,2}}\right| < 4.7, \quad \eta_{j_1}\eta_{j_2} < 0,
$$
$$
N_\ell = 0, \quad N_j^{\text{central}} = 0, \quad \Delta\phi\left(\vec{p}_{T,j_1}, \vec{p}_{T,j_2}\right) < 2.2, \quad \Delta\phi\left(\vec{p}_{T,\text{miss}}, \vec{p}_{T,j}\right) > 0.5.
$$

(32)

Here the number of light-flavoured jets and charged leptons is denoted by $N_j$ and $N_\ell$, respectively, $j_1$ and $j_2$ indicate the two jets with the largest $p_T$, $\Delta\phi$ denotes the azimuthal angular separation and $\vec{p}_{T,\text{miss}}$ is the vector sum of the transverse momenta of all invisible particles. The implementation of the lepton veto is identical to the one in the CMS analysis [68] and the central jet veto applies to additional jets with $p_{T,j} > 30\,\text{GeV}$ with $\min(\eta_{j_{1,2}}) < \eta_j < \max(\eta_{j_{1,2}})$ and the $\Delta\phi\left(\vec{p}_{T,\text{miss}}, \vec{p}_{T,j}\right)$ cut affects any jet with $p_{T,j} > 30\,\text{GeV}$. We use the CMS card of `Delphes 3` for the fast detector simulation in our HL-LHC analysis.

The normalised distributions of the variables that provide the main handles to reject the SM backgrounds are shown in Figure 8. In the case of $E_{T,\text{miss}}$ we impose in our HL-LHC analysis $E_{T,\text{miss}} > 180\,\text{GeV}$ following the ATLAS search [67]. As can be seen from the upper left panel in the figure, the signal would prefer a milder cut on $E_{T,\text{miss}}$, however, we rely on $E_{T,\text{miss}}$ as a trigger and the trigger efficiency drops dramatically for lower values of $E_{T,\text{miss}}$. Notice that the shape of the $E_{T,\text{miss}}$ distribution of the signal is very similar to that of the dominant EW background. To separate the signal from the EW background we thus consider the invariant mass $m_{j_1 j_2}$ and the pseudorapidity separation $\Delta\eta_{j_1 j_2}$ of the two tagging jets characteristic for VBF-like signatures. The corresponding distributions are shown in the upper right and the lower panel of Figure 8, respectively. In order to exploit the discriminating power of $\Delta\eta_{j_1 j_2}$ we require $\Delta\eta_{j_1 j_2} > 4.2$ in our HL-LHC analysis and then optimise the $m_{j_1 j_2}$ cut such that the significance is maximised, separately for each $m_\psi$ hypothesis. Notice in this context that the differential $pp \to 2j + E_{T,\text{miss}}$ cross sections in the fermionic Higgs-portal model are all proportional to $N_\psi c_\psi^2/f^2$ meaning that only the mass $m_\psi$ of the new fermions changes the shape

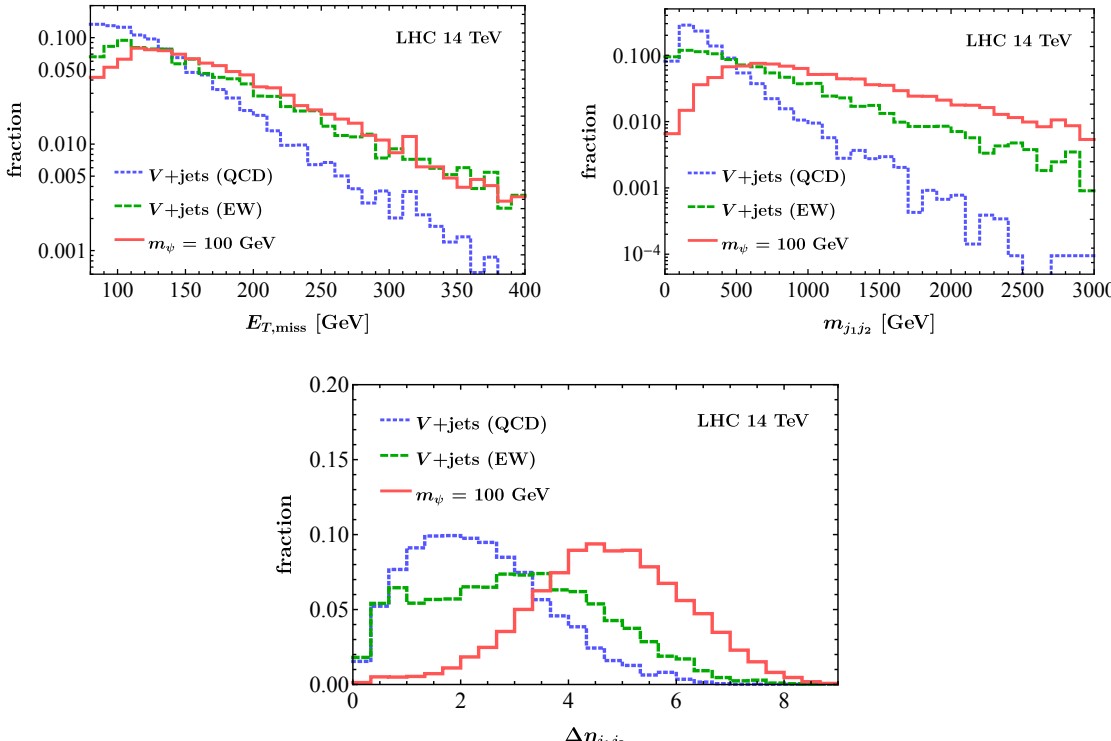

Figure 8: Normalised $E_{T,\text{miss}}$ (upper left), $m_{j_1 j_2}$ (upper right) and $\Delta\eta_{j_1 j_2}$ (lower) distributions for LHC collisions at a CM energy of $\sqrt{s} = 14\,\text{TeV}$. The QCD $V$+jets (dashed blue) and EW $V$ + jets (dotted green) backgrounds and the fermionic Higgs-portal signal with $m_\psi = 100\,\text{GeV}$ (solid red) are shown. Except for the $\Delta\eta_{j_1 j_2}$ spectrum all distributions are subject to the basic selections (32). Additional details can be found in the main text.

of the relevant kinematic distributions. After all cuts we typically end up with a sub-percent signal-to-background ratio implying that the analysis is systematics limited. In Section 4 where we study the collider reach of the VBF Higgs production channel, we will make different assumptions about the systematic uncertainties, following the methodology of the CERN Yellow report on the HL-LHC physics potential [72].

## 4 Collider reach: Twin Higgs, $f = 3v$

Figure 9 provides a comparison of the HL-LHC reach of different search strategies in the $m_\psi - |c_\psi|$ plane, considering the full integrated luminosity of $3\,\text{ab}^{-1}$ at $\sqrt{s} = 14\,\text{TeV}$. We make the assumptions $N_\psi = 3$, $f = 3v$, $\mu_* = 4\pi f$ and $C_6(\mu_*) = C_{H\square}(\mu_*) = 0$. In Appendix D we consider different non-zero initial conditions $C_6(\mu_*)$ and $C_{H\square}(\mu_*)$. Notice that the chosen $f$ corresponds to the minimal value in twin-Higgs models that is consistent with existing experimental measurement, necessitating $v/f \lesssim 1/3$ [73]. Since the tuning amounts to roughly $2v^2/f^2$ in minimal twin-Higgs models [38, 74] the choice of $f = 3v$ implies only a very modest tuning of around 20% — we repeat our study for a more tuned twin-Higgs model in Appendix B. Notice that the used parameters correspond to a UV completion that becomes strongly-coupled at $\Lambda \simeq 4\pi f \simeq 9\,\text{TeV}$, a scale that is well above the energies probed by the shown LHC search strategies. The solid blue line represents the 95% CL limits obtained

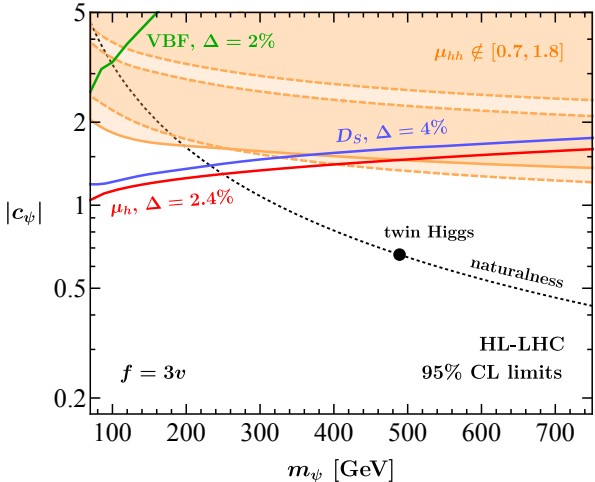

Figure 9: HL-LHC reach of different search strategies in the $m_\psi - |c_\psi|$ plane corresponding to the assumptions $N_\psi = 3$, $f = 3v$, $\mu_* = 4\pi f$ and $C_6(\mu_*) = C_{H\square}(\mu_*) = 0$. The solid blue, solid green and solid red lines represent the 95% CL limits derived from our $D_S$ analysis, our study of the VBF Higgs production channel and a hypothetical measurement of the global Higgs signal strength $\mu_h$, respectively. The assumed systematic uncertainties or accuracies, when applicable, are indicated. Regions above the coloured lines are disfavoured. The region bound by the solid (dashed) orange line arises from imposing that the signal strength in double-Higgs production satisfies $\mu_{hh} \notin [0.7, 1.8]$ for $c_\psi > 0$ ($c_\psi < 0$). The naturalness bound (2) is represented by the dotted black line, while the point $\{m_\psi, |c_\psi|\} = \{y_t f / \sqrt{2}, y_t / \sqrt{2}\} \simeq \{490\,\text{GeV}, 0.7\}$, corresponding to (2) for the special case of a standard twin-Higgs model, is displayed as a black dot. More details are provided in the main text.

from a binned-likelihood analysis of the ME-based kinematic discriminant (30), following the methodology of [27], with a systematic uncertainty assumed to be $\Delta = 4\%$. Conversely, the solid green line indicates the constraint derived from our investigation of off-shell Higgs-boson production in the VBF channel. Like in [72] our VBF analysis assumes a systematic uncertainty of $\Delta = 2\%$. For details on the statistical analyses see [15, 19, 27]. HL-LHC measurements of the global Higgs signal strength $\mu_h$ are anticipated to achieve an accuracy of $\Delta = 2.4\%$, accounting for the anticipated total systematic uncertainties [75]. Utilising the quoted precision together with $\mu_h = 1 + \delta Z_h$ and (16) leads at 95% CL to the solid red line. We note that the employed systematic uncertainties correspond to those called S2 in the CERN Yellow report on the HL-LHC physics potential [72]. In Appendix A we present results for the less optimistic scenario S1, which assumes that the HL-LHC systematic uncertainties are equal to those in LHC Run 2. The 95% CL bound $\kappa_\lambda \in [0.18, 3.6]$ on the modifications $\kappa_\lambda = \lambda/\lambda_{\text{SM}}$ of the trilinear Higgs coupling, as determined by the CMS projection [62], translates to $\mu_{hh} \in [0.7, 1.8]$ for the signal strength in $pp \to hh$ production at the HL-LHC. Based on our full one-loop calculation of double-Higgs production in the theory described by (3), we derive the solid and dashed orange lines corresponding to $c_\psi > 0$ and $c_\psi < 0$, respectively. Finally, the dashed black line represents the naturalness condition (2) while the black dot indicates the point $\{m_\psi, |c_\psi|\} = \{y_t f / \sqrt{2}, y_t / \sqrt{2}\} \simeq \{490\,\text{GeV}, 0.7\}$, which corresponds to the naturalness bound (2) in the special case of a standard twin Higgs model.

A comparison of the 95% CL constraints displayed in Figure 9 shows that in twin-Higgs realisations of (3) the indirect probes, i.e. $D_S$, $\mu_{hh}$ and $\mu_h$, provide notably more stringent

bounds in the $m_\psi - |c_\psi|$ plane than the direct search for $\psi$ pair production in the VBF Higgs production channel. This feature is a consequence of the higher-dimensional nature of (1) that leads to large logarithmic effects that are associated to the RG evolution of $Q_6$ and $Q_{H\square}$ — cf. (5) and (6). These operators enter the quantum version (3) of the fermionic Higgs-portal Lagrangian and are needed to renormalise (1) at the one-loop level, and thus are necessary for a consistent computation of loop-induced processes such as $gg \to h^* \to ZZ$ or $gg \to hh$. Tree-level processes like $q\bar{q}' \to q\bar{q}'h^* \to q\bar{q}'\psi\bar{\psi}$ can, on the other hand, be consistently calculated using (1) and as a result, at the Born level, they remain insensitive to the logarithmically enhanced effects associated to the quantum structure of the theory. It is also evident from the figure that while the constraints arising from $pp \to h^* \to ZZ$ and $pp \to h$ decouple slowly with increasing $m_\psi$, the $pp \to hh$ limits tend to become stronger for larger $m_\psi$ values. This behaviour is readily understood by noticing that while (16) and (21) contain logarithms of the form $c_\psi^2 \ln\left(\mu_*^2/m_\psi^2\right)$, the correction (28) entails terms that scale as $c_\psi^3 (m_\psi/f) \ln\left(\mu_*^2/m_\psi^2\right)$. Notice that which indirect constraint provides the best bound depends on both the value of $m_\psi$ and the sign of $c_\psi$. In the case of $c_\psi > 0$ the limit from $D_S(\mu_h)$ turns out to be stronger for $m_\psi \lesssim 360\,\text{GeV}$ ($m_\psi \lesssim 500\,\text{GeV}$) while for larger masses the observable $\mu_{hh}$ represents the best constraint. For $c_\psi < 0$ the corresponding limits are $m_\psi \lesssim 340\,\text{GeV}$ and $m_\psi \lesssim 400\,\text{GeV}$, respectively. The dependence of the $gg \to hh$ amplitude on $c_\psi$ and $m_\psi$ furthermore leads for $c_\psi < 0$ to a funnel in the $m_\psi - |c_\psi|$ plane which cannot be excluded by double-Higgs production because the signal strength $\mu_{hh}$ is SM-like — for details see the discussion in Section 3.2. Notice finally that all constraints shown in Figure 9 depend in a non-negligible way on the assumed systematic uncertainties or accuracies. In view of these caveats one can conclude that to fully exploit the HL-LHC potential in probing fermionic Higgs-portal interactions of the form (1) one should consider all indirect and direct probes displayed in the figure. If this is done one sees that it should be possible to explore fermionic Higgs-portal models (1) that are compatible with the naturalness bound (2) for fermion masses in the range of $m_\psi \in [62.5, 250]\,\text{GeV}$. Unfortunately, this implies that the point $\{m_\psi, |c_\psi|\} = \{y_t f/\sqrt{2}, y_t/\sqrt{2}\} \simeq \{490\,\text{GeV}, 0.7\}$, representing a natural standard twin-Higgs model with $f = 3v$, cannot be probed at the HL-LHC.

In our HE-LHC (FCC) study, we consider $pp$ collisions at $\sqrt{s} = 27\,\text{TeV}$ ($\sqrt{s} = 100\,\text{TeV}$) and an integrated luminosity of $15\,\text{ab}^{-1}$ ($30\,\text{ab}^{-1}$). While we expand the $m_{4\ell}$ window to $1000\,\text{GeV}$ ($1500\,\text{GeV}$) at the HE-LHC (FCC), the selection cuts and detection efficiencies in our analyses mirror those outlined in Section 2.4. Technical improvements in the HE-LHC and FCC detectors, such as extended pseudorapidity coverages [76, 77], which could enhance the reach of the off-shell Higgs-boson production channel, are not considered in what follows. Additionally, we use the values of the $K$-factors provided in Section 2.4, obtained for LHC collisions, to calculate QCD-improved predictions for the kinematic variable $D_S$ à la (31). Given that the assumed systematic uncertainties play a significant role in determining the HE-LHC and FCC reach for constraining fermionic Higgs-portal interactions of the form (1), we consider these simplifications fully justified. In our HE-LHC analysis of VBF off-shell Higgs production the baseline cuts (32) remain unchanged with the exception that we now require $|\eta_{j_{1,2}}| < 4.9$. We additionally increase the missing transverse energy cut to $E_{T,\text{miss}} > 200\,\text{GeV}$. The selections imposed in the FCC VBF analysis resemble those used at the HE-LHC apart from that we allow for tagging jets up to $|\eta_{j_{1,2}}| < 6$. Our fast detector simulation at the HE-LHC (FCC) employs the HL-LHC (FCC) `Delphes 3` card.

The HE-LHC and FCC sensitivities of various search strategies for the fermionic Higgs-portal coupling (1) are presented in the two panels of Figure 10. Similar to our HL-LHC study, different systematic uncertainties are assumed for individual search channels. For our $gg \to h^* \to ZZ$ analysis, we consider $\Delta = 2\%$ and $\Delta = 1\%$ as the systematic uncertainties at the HE-LHC and FCC, respectively. Given a target precision at the FCC of 1.8% in the

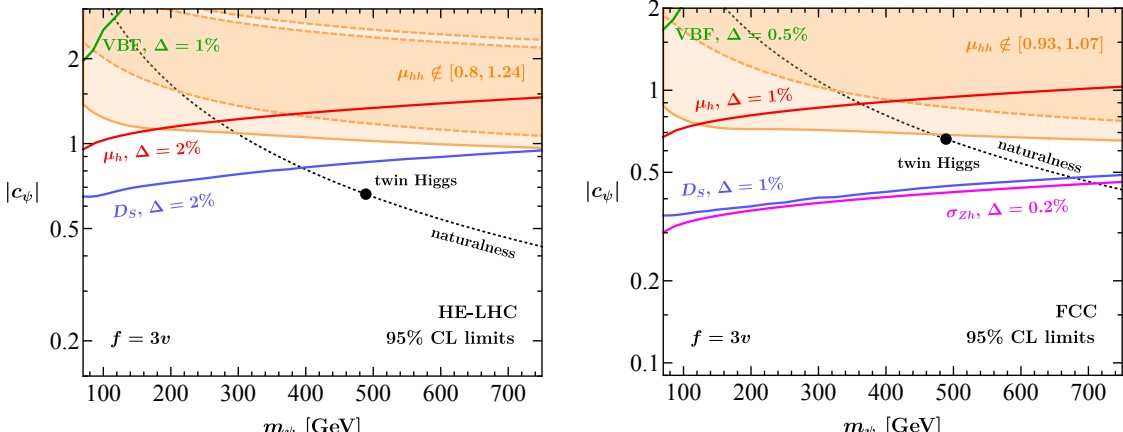

Figure 10: Projected reach of different search strategies in the $m_\psi - |c_\psi|$ plane for the HE-LHC (left panel) and FCC (right panel). The presented constraints are based on $N_\psi = 3$, $f = 3v$, $\mu_* = 4\pi f$ and $C_6(\mu_*) = C_{H\square}(\mu_*) = 0$. In addition to the constraints illustrated in Figure 9, the FCC case features an additional solid magenta line representing the 95% CL limit derived from a precision measurement of the $Zh$ production cross section ($\sigma_{Zh}$). The colour scheme and interpretation of the remaining constraints mirror those presented in the previous figure. Consult the main text for additional explanations.

$pp \to ZZ \to 4\ell$ channel [78], and considering potential advancements in theory and experiment, a final systematic uncertainty of 2% (1%) at the HE-LHC (FCC) appears conceivable. Regarding the global Higgs signal strength $\mu_h$, we employ $\Delta = 2\%$ and $\Delta = 1\%$ [78] at the HE-LHC and FCC, respectively. The anticipated 95% CL bounds on modifications of the trilinear Higgs coupling at these two colliders are expected to be $\kappa_\lambda \in [0.7, 1.3]$ and $\kappa_\lambda \in [0.9, 1.1]$ — see for example [72, 79, 80] for detailed discussions. These constraints translate into limits of $\mu_{hh} \in [0.80, 1.24]$ and $\mu_{hh} \in [0.93, 1.07]$ on the signal strength in double-Higgs production. In the case of the VBF off-shell Higgs production study we instead employ $\Delta = 1\%$ and $\Delta = 0.5\%$ at the HE-LHC and the FCC, respectively. In Appendix A we present HE-LHC and FCC projections for $pp \to h^* \to ZZ$ and $pp \to 2jh^* \to 2j\psi\bar{\psi}$, making less optimistic assumptions about the future systematic uncertainties of these two search channels.

The first take-home message from Figure 10 is that the reach of the studied direct search strategy is in general not competitive with the considered indirect tests. The basic reason is again that the loop-induced processes receive logarithmically enhanced effects from $Q_6$ and $Q_{H\square}$ which are absent in all tree-level transitions that only probe (1) directly. Another interesting feature that one observes from Figure 10 is that for the same systematic uncertainties the constraints arising from the off-shell Higgs-boson measurements are more stringent than the limits that follow from the on-shell Higgs-boson signal strength. The enhanced sensitivity from $D_S$ compared to $\mu_h$ originates from the fact that the former observable is sensitive to events that lie in the tails of the kinematic distributions, while such events have only a limited weight in the total Higgs production cross sections. To further illustrate the latter feature we show in the case of the FCC the exclusion that follows from an extraction of the signal strength in $e^+e^- \to Zh$ with an accuracy of $\Delta = 0.2\%$ as a solid magenta line. It is conceivable that an electron-positron ($e^+e^-$) precursor of the FCC, operating at a CM energy of $\sqrt{s} = 240\,\text{GeV}$ with an integrated luminosity of $5\,\text{ab}^{-1}$ [81], could achieve this accuracy. From the right panel in Figure 10 it is evident that only such a precision $e^+e^-$ measurement would allow to set stronger bounds than the hypothetical FCC measurement of off-shell Higgs production in $pp$ collisions considered by us. Notice finally that in the case of $f = 3v$ the HE-LHC (FCC) is

likely to be able to provide coverage of the $|c_\psi|$ values of natural fermionic Higgs-portal models of the form (1) for $m_\psi \in [62.5, 400]$ GeV ($m_\psi \in [62.5, 700]$ GeV). As a result, only the FCC is expected to allow to test the point $\{m_\psi, |c_\psi|\} = \{y_t f/\sqrt{2}, y_t/\sqrt{2}\} \simeq \{490\,\text{GeV}, 0.7\}$, which represents the case of a natural standard twin-Higgs model with $f = 3v$. Further studies of the collider reach employing two different choices for $N_\psi$, $f$ and $\mu_*$ are presented in Appendix B and Appendix C, respectively.

## 5 Conclusions

In this article, we have investigated the extent to which upcoming measurements at hadron colliders can detect and limit the strength of the fermionic Higgs portal (1). Our focus lies on scenarios where the new fermions cannot be observed through exotic decays of the 125 GeV Higgs boson. These portals emerge in models of neutral naturalness such as twin-Higgs scenarios as well as in Higgs-portal DM models, presenting significant challenges for detections at colliders. To keep the discussion as model-independent as possible, we worked in the context of an EFT in which the fermionic Higgs portal is augmented by the dimension-six operators $Q_6$ and $Q_{H\square}$ — see (3) and (4). As we have explained in detail, once radiative corrections are considered in the context of (1), the resulting quantum theory unavoidably contains $Q_6$ and $Q_{H\square}$. In fact, at the one-loop level and up to dimension six, only the latter two operators emerge, making (3) the minimal EFT that allows to consistently calculate off-shell and double-Higgs production in fermionic Higgs-portal models.

Our one-loop computations of the $gg \to h^* \to ZZ \to 4\ell$ and $gg \to hh$ processes in the context of (3) have been implemented into MCFM 8.0. This Monte Carlo generator has then been used to study the reach of the HL-LHC, the HE-LHC and the FCC in constraining the fermionic Higgs portal via off-shell and double-Higgs production. We found that in the fermionic Higgs-portal scenario, the high-energy tails of relevant kinematic distributions in both off-shell and double-Higgs production are enhanced. This enhancement stems from the higher-dimensional nature of (1), necessitating the inclusion of dimension-six operators in the computation of $gg \to h^* \to ZZ$ and $gg \to hh$ amplitudes to ensure UV finiteness. Consequently, the RG flow of the corresponding Wilson coefficients induces logarithmically enhanced corrections, altering the $gg \to h^* \to ZZ$ and $gg \to hh$ matrix elements and thereby influencing the resulting kinematic distributions in a non-trivial manner. As a result, quantum enhanced indirect probes such as off-shell and double-Higgs production in general provide a better sensitivity to interactions of the form (1) than direct probes like VBF Higgs production. This observation holds particularly true for low-energy realisations of (1), which originate from theories like twin-Higgs models that are characterised by a strong-coupling UV regime. The sensitivity of off-shell and double-Higgs production to the initial conditions of the Wilson coefficients of $Q_6$ and $Q_{H\square}$ is discussed in Appendix D.

Before drawing the curtain let us compare the main findings obtained in this work to the global picture of constraints that emerges in the case of the marginal Higgs-portal model [11, 12, 15, 19, 24–27]. The main difference between the collider phenomenology in the fermionic and the marginal Higgs-portal model is that in the latter BSM model the direct constraints that arise from processes such as $pp \to 2jh^* \to 2j\psi\bar\psi$ and $pp \to t\bar t h^* \to t\bar t \psi\bar\psi$ are often more important than those that stem from indirect tests associated to loop processes such as $pp \to h^* \to ZZ \to 4\ell$ or $pp \to hh$. This is a simple consequence of the fact that unlike (1) the marginal Higgs-portal model can be renormalised without the need to introduce higher-dimensional operators. The large logarithmic effects that appear in the calculation of Higgs-boson observables in the context of the fermionic Higgs portal are therefore not present in the marginal case. Notice that on general grounds, the logarithmic enhancement of loop effects is

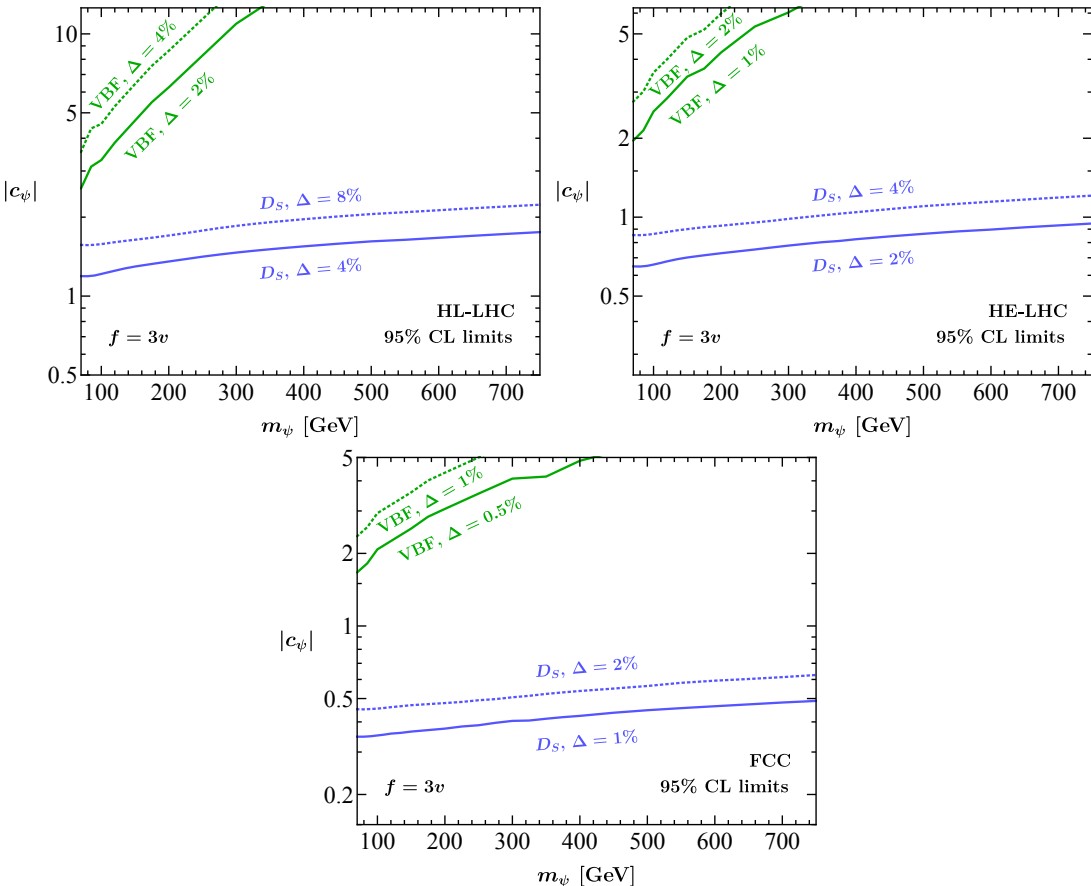

Figure 11: Reach of off-shell (blue) and VBF (green) Higgs production in the $m_\psi - |c_\psi|$ plane for the HL-LHC (upper left panel), HE-LHC (upper right panel) and FCC (lower panel). The shown results are based on $N_\psi = 3$, $f = 3v$, $\mu_* = 4\pi f$ and $C_6(\mu_*) = C_{H\square}(\mu_*) = 0$. The assumed systematic uncertainties are indicated in each case.

also expected in the case of the vector or kinetic Higgs-portal model. We leave studies of the loop-induced collider phenomenology in these models for future research.

## Acknowledgments

We thank Ennio Salvioni for collaboration in the initial stage of this project. His useful comments concerning the first version of this manuscript are also acknowledged.

**Funding information** MR is supported in part by the NSF grant PHY-2014071 and by a Feodor-Lynen Research Fellowship awarded by the Humboldt Foundation. The work of KS and AW has been partially supported by the DFG Cluster of Excellence 2094 ORIGINS, the Collaborative Research Center SFB1258 and the BMBF grant 05H18WOCA1.

# A  Systematic uncertainties

In this appendix, we examine how different assumptions on the systematic uncertainties in the off-shell and VBF Higgs production channels affect the constraints on the parameter space of the fermionic Higgs-portal model (1). The fact that the assumptions on the systematic uncertainties $\Delta$ play a crucial role in constraining the $m_\psi - |c_\psi|$ parameter space using both off-shell and VBF Higgs production is illustrated in Figure 11. The shown results are based on $N_\psi = 3$, $f = 3v$, $\mu_* = 4\pi f$ and $C_6(\mu_*) = C_{H\Box}(\mu_*) = 0$, i.e. the very same parameters studied before in Section 4. In the case of the HL-LHC the employed systematic uncertainties correspond to the scenarios called S1 and S2 in the CERN Yellow report on the HL-LHC physics potential [72], while for what concerns the HE-LHC and FCC we rely on the FCC physics opportunities study [78]. The first observation is that the gain in sensitivity when halving the assumed systematic uncertainties is to first approximation independent of the mass $m_\psi$ for both the $D_S$ analysis of $pp \to ZZ \to 4\ell$ as well as our $pp \to 2j\psi\bar{\psi}$ search strategy. Second, the impact of reducing the systematic uncertainties is also largely independent of the considered collider. Numerically, halving $\Delta$ leads to relative improvements in the bounds on $|c_\psi|$ of around 20% and 30% in the case of off-shell and VBF Higgs production, respectively. Finally, it is worth noting that the constraints on the $m_\psi - |c_\psi|$ plane derived from off-shell and VBF Higgs production both exhibit a noticeable sensitivity to the assumed systematic uncertainties. However, regardless of the uncertainty scenario considered, the limits derived from $pp \to ZZ \to 4\ell$ are consistently stronger than those originating from $pp \to 2j\psi\bar{\psi}$.

# B  Collider reach: Twin Higgs, $f = 6v$

In Section 4, we have performed a comprehensive study of the collider reach for the fermionic Higgs-portal model (1) assuming parameters that are appropriate to capture the phenomenology of a twin-Higgs model with a low value of $f$ and a very modest tuning of about 20%. Below we repeat our analysis for the choices $N_\psi = 3$, $f = 6v$, $\mu_* = 4\pi f$ and $C_6(\mu_*) = C_{H\Box}(\mu_*) = 0$. These parameters correspond to a twin-Higgs model with a tuning of about $2v^2/f^2 \simeq 5\%$. Note that the parameters employed describe a UV completion, which becomes strongly coupled at $\Lambda \simeq 4\pi f \simeq 18\,\text{TeV}$, a scale significantly higher than the energies probed by the relevant LHC search strategies. Our projections of the HL-LHC (upper left panel), HE-LHC (upper right panel) and FCC (lower panel) reach are summarised in Figure 12. Compared to the results shown in Figure 9 and Figure 10 that employ $f = 3v$, one observes that for $f = 6v$ the parameter region in the $m_\psi - |c_\psi|$ plane that can be explored at a given hadron collider is significantly reduced. In fact, in the case of the VBF limits the bounds on $|c_\psi|$ are weaker by a factor of 2, while in the case of $D_S$, $\mu_h$ and $\sigma_{Zh}$ the constraints are less stringent by a factor of around 1.8. This difference is easy to understand qualitatively by noticing that the former observable scales as $c_\psi^2 \, (v^2/f^2)$ while the latter observables involve terms of the form $c_\psi^2 \, (v^2/f^2) \ln\left(\mu_*^2/m_\psi^2\right)$. Another feature that is clearly visible in all panels is that the constraints from double-Higgs production become more stringent with increasing fermion mass $m_\psi$. This behaviour is again a result of (28) containing terms that scale as $c_\psi^3 \, (m_\psi/f) \ln\left(\mu_*^2/m_\psi^2\right)$ which provide the dominant contribution to $pp \to hh$ production if $m_\psi$ approaches $f$. Notice furthermore that the obtained direct limits are not competitive with the indirect constraints. The three panels also show that for $f = 6v$ it should be possible to explore fermionic Higgs-portal models (1) that are compatible with the naturalness bound (2) for fermion masses in the range of $m_\psi \in [62.5, 300]\,\text{GeV}$, $m_\psi \in [62.5, 420]\,\text{GeV}$ and $m_\psi \in [62.5, 800]\,\text{GeV}$ at the HL-LHC, the HE-LHC and the FCC, respectively. These results imply that even the FCC will probably not be able to test the point $\{m_\psi, |c_\psi|\} = \{y_t f/\sqrt{2}, y_t/\sqrt{2}\} \simeq \{980\,\text{GeV}, 0.7\}$, which represents the

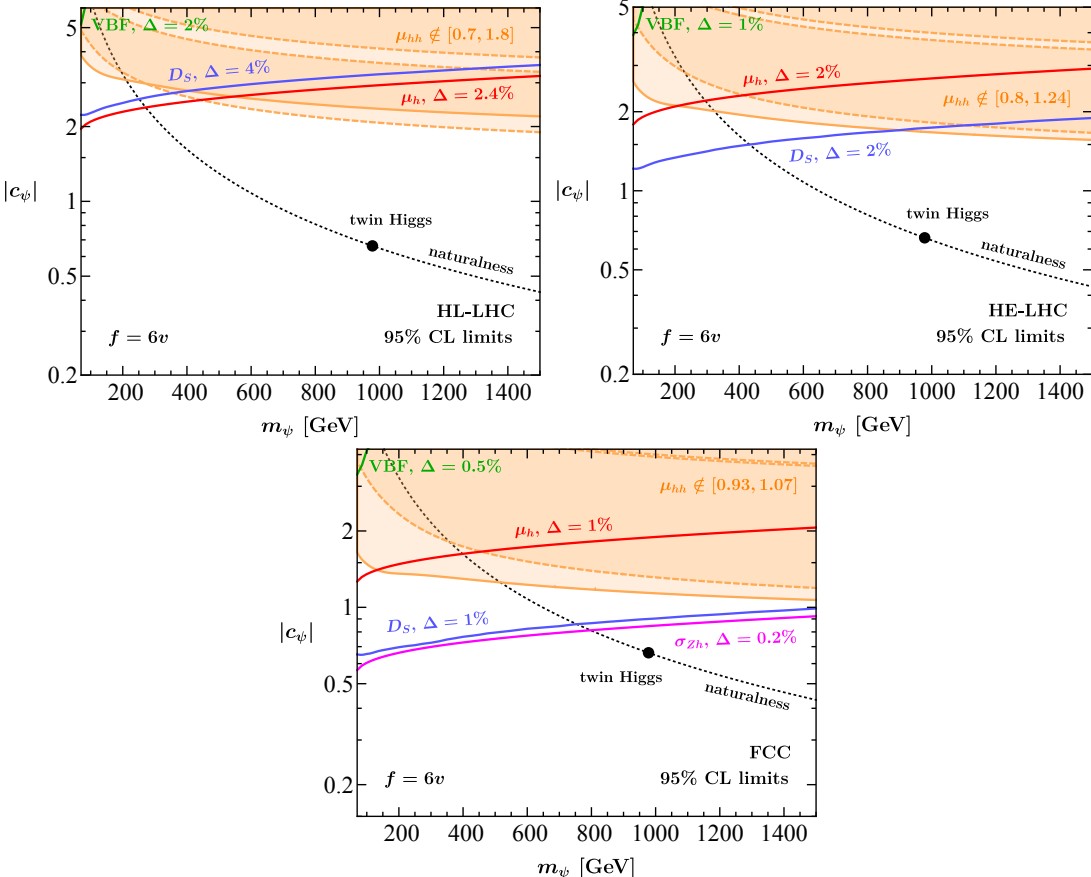

Figure 12: As Figure 9 and Figure 10 but assuming $N_\psi = 3$, $f = 6v$, $\mu_* = 4\pi f$ and $C_6(\mu_*) = C_{H\square}(\mu_*) = 0$. Notice that all constraints following from VBF off-shell Higgs production include only statistical uncertainties. Additional details can be found in the main text.

case of a natural standard twin-Higgs model with $f = 6v$. In this context, it is important to recall that our analysis of double-Higgs production depends entirely on the signal strength of the fully inclusive cross section. By performing measurements of differential distributions in $pp \to hh$, such as the invariant di-Higgs mass $m_{hh}$, it might be possible to improve the bounds derived from double-Higgs production. However, such an analysis is beyond the scope of our article.

## C  Collider reach: DM Higgs portal

In Section 4 and Appendix B, we have studied two realisations of (1) employing choices of $N_\psi$, $f$ and $\mu_*$ that allow to model the dynamics of standard twin-Higgs scenarios. In both cases we have made the choices $\mu_* = 4\pi f$ and $C_6(\mu_*) = C_{H\square}(\mu_*) = 0$. We now give up on the assumption that the UV cut-off scale $\Lambda$ is larger than $f$ by a loop factor as naturally expected in strongly-coupled theories. For weakly-coupled models one instead expects $\Lambda \simeq f$, and therefore we choose $f = 1\,\text{TeV}$, $\mu_* = f$ and $C_6(\mu_*) = C_{H\square}(\mu_*) = 0$ to model this case. In order to make contact with the fermionic Higgs-portal models considered in invisible Higgs-boson decays at the LHC — see [82,83] for the latest ATLAS and CMS searches of this kind — we furthermore employ $N_\psi = 1$. The HL-LHC (upper left panel), HE-LHC (upper right panel) and

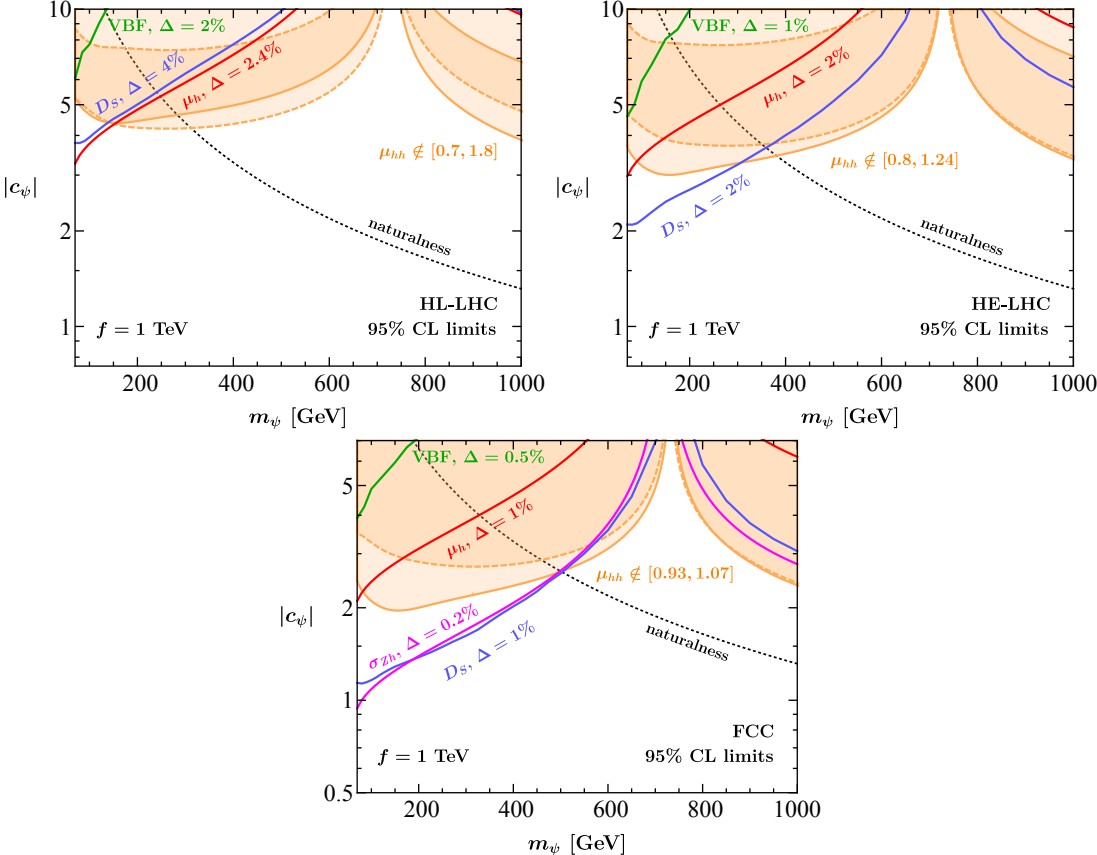

Figure 13: As Figure 12 but with $N_\psi = 1$, $f = 1\,\text{TeV}$, $\mu_* = f$ and $C_6(\mu_*) = C_{H\square}(\mu_*) = 0$. The assumed systematic uncertainties or accuracies are indicated. See the main text for more explanations.

FCC (lower panel) projections corresponding to the above parameter choices are displayed in Figure 13. Compared to the results shown in Figures 9, 10 and 12 one obvious difference is the shape of the $D_S$, $\mu_h$, $\sigma_{Zh}$ and $\mu_{hh}$ constraints. While in the studied twin-Higgs scenarios these observables are fully dominated by the logarithmically enhanced corrections of the form $c_\psi^2\,(v^2/f^2)\ln\left(\mu_*^2/m_\psi^2\right)$ or $c_\psi^3\,(m_\psi/f)\ln\left(\mu_*^2/m_\psi^2\right)$ this is not the case in the considered DM Higgs-portal model because the UV cut-off scale is only taken to be $\mu_* = f$ and not $\mu_* = 4\pi f$. In fact, for the parameter choices $N_\psi = 1$, $f = 1\,\text{TeV}$, $\mu_* = f$ and $C_6(\mu_*) = C_{H\square}(\mu_*) = 0$ the individual terms in (16), (21) and (28) tend to cancel for $m_\psi \simeq 700\,\text{GeV}$, and this explains why the $D_S$, $\mu_h$, $\sigma_{Zh}$ and $\mu_{hh}$ constraints weaken notably for $\psi$ masses in this vicinity. It is also evident from all three panels that the obtained direct limits are in general weaker than the constraints that stem from the studied quantum enhanced indirect probes. One furthermore sees that fermionic DM Higgs-portal models (1) that comply with the naturalness condition (2) can be explored for $m_\psi \in [62.5, 300]\,\text{GeV}$, $m_\psi \in [62.5, 400]\,\text{GeV}$ and $m_\psi \in [62.5, 500]\,\text{GeV}$ at the HL-LHC, the HE-LHC and the FCC, respectively. Given the complementary of the indirect constraints shown in the three panels of Figure 13, it should be clear that in order to exploit the full potential of the HL-LHC, HE-LHC and FCC in testing fermionic Higgs-portal interactions requires a multi-prong approach that consists in studying various Higgs observables such as $D_S$, $\mu_h$, $\sigma_{Zh}$ and $\mu_{hh}$.

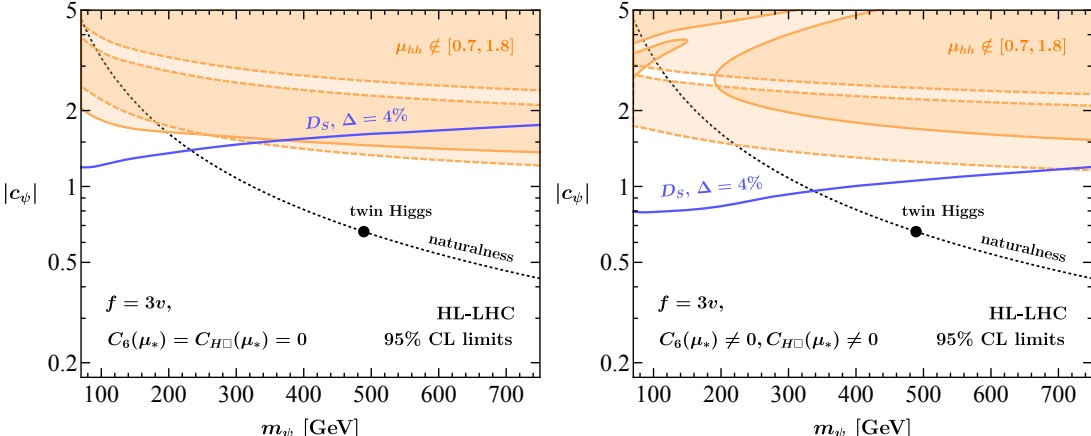

Figure 14: HL-LHC reach of the variable $D_S$ and the signal strength $\mu_{hh}$. All displayed constraints utilise $N_\psi = 3$, $f = 3v$ and $\mu_* = 4\pi f$. In the left panel, we furthermore assume that $C_6(\mu_*) = C_{H\square}(\mu_*) = 0$, while in the right panel, we instead employ the initial conditions (D.1). Consult the main text for more details.

## D  Model dependence in twin-Higgs models

Until now we have always assumed that the Wilson coefficients of the operators $Q_6$ and $Q_{H\square}$ introduced in (4) vanish identically at the high-energy matching scale $\mu_*$, meaning that we have always taken $C_6(\mu_*) = C_{H\square}(\mu_*) = 0$ in our numerical analyses. In order to study the model dependence of our twin-Higgs predictions, we are now abandoning this assumption.

In the case that the interactions (3) arise from the UV embedding into a minimal twin-Higgs model, one can calculate the initial conditions $C_6(\mu_*)$ and $C_{H\square}(\mu_*)$. Including all tree-level effects as well as the loop corrections that are enhanced by either the top-quark Yukawa coupling $y_t$ or the fermionic Higgs-portal coupling $c_\psi$, we find

$$C_6(\mu_*) = \frac{1}{f^2}\left[\frac{2m_h^2}{3v^2} + \frac{y_t^4 - 2c_\psi^4}{4\pi^2}\right], \qquad C_{H\square}(\mu_*) = \frac{1}{2f^2}. \tag{D.1}$$

Note that in the above initial conditions, we have separated the contribution from top-quark loops, which are proportional to $y_t^4$, and top-quark partner loops, which are proportional to $\hat{y}_t^4 = 4c_\psi^4$, to the effective potential. In a natural twin-Higgs model, one has $\hat{y}_t = y_t$. However, since $c_\psi = -\hat{y}_t/\sqrt{2}$, a modification of $c_\psi$ directly translates into a modification of $\hat{y}_t$, which in turn also affects the contribution to the effective potential from top-quark partner loops.

In Figure 14 we repeat the analysis performed at the beginning of Section 4, utilising the initial conditons (D.1) instead of $C_6(\mu_*) = C_{H\square}(\mu_*) = 0$. The rest of the input parameters agrees with those used in Figure 9. In the case of the variable $D_S$, one sees that the bound that is obtained using the non-zero initial conditions given in (D.1) is stronger than the one that derives from $C_6(\mu_*) = C_{H\square}(\mu_*) = 0$. In fact, this feature is readily understood by noting that the two terms in (22) interfere constructively for all UV realisations of (3) that predict $C_{H\square}(\mu_*) > 0$. In twin-Higgs models, the positivity of the initial condition $C_{H\square}(\mu_*)$ is a model-independent prediction. Consequently, setting $C_{H\square}(\mu_*) = 0$ yields $D_S$ limits that serve as conservative upper bounds on $|c_\psi|$, expected to apply broadly across twin-Higgs scenarios. In the case of the constraints that follow from the signal strength $\mu_{hh}$, it is evident from Figure 14 that for $c_\psi < 0$ the bounds that are obtained for $C_6(\mu_*) = C_{H\square}(\mu_*) = 0$ and (D.1) are very similar. For $c_\psi > 0$ one instead observes that the exclusions are notable different. The

difference is due to the interplay of the $c_\psi^3$ terms in (28) and the $c_\psi^4$ terms in (D.1), which tend to cancel for certain values of $m_\psi$, leading to funnels in the $m_\psi - |c_\psi|$ plane. Based on the aforementioned findings, we argue that setting the initial conditions $C_6(\mu_*)$ and $C_{H\square}(\mu_*)$ to zero as done in Section 4 is justified. This approach offers conservative and relatively model-independent constraints applicable to general twin-Higgs models.

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
