# Peer review of "Quantum collider probes of the fermionic Higgs portal"

_SciPost Physics, doi:SciPost Phys. 16, 112 (2024)_

## Round 1 · Referee Report · Anonymous · 2024-1-7

Report
The present manuscript investigates how upcoming hadron collider experiments can constrain the fermionic Higgs portal, specifically when new fermions are not directly observed in exotic Higgs decays. The study assesses the reach of the HL-LHC, HE-LHC, and FCC for off-shell Higgs and double-Higgs production. Their findings show that quantum-enhanced indirect probes offer the best sensitivity. While the paper is worthy of publication, certain aspects would benefit further refinement for publication.
Requested changes
1) Please comment on the validity of the EFT analysis of Eq. 1. In particular, I observed that Eq. 26 introduces terms scaling as $c_\psi^6$ at the cross-sectional level for double Higgs production. It appears that these terms dominate the new physics contributions rather than the lower-dimensional ones.
2) In this context, what is the rationale behind including only the two dim-6 operators in Eq. 4? Considering that the final calculation accounts for significantly higher-order terms in the EFT expansion, what motivates the specific choice to limit the inclusion to these two operators?
3) Considering the chosen cutoff for the theory (f=3v), there seems to be a lack of justification for the upper range of the $m_{4l}$ analysis, specifically including events up to $m_{4l}=1$ TeV for the HE-LHC and $m_{4l}=1.5$ TeV for the FCC.
4) On page 11: "The inclusion of higher-order QCD corrections furthermore reduces the scale uncertainties of DS by a factor of about 3 from around 7.5% to 2.5%." Please provide more details on how these scale uncertainties are derived.
5) Regarding Fig. 5, it would be beneficial to assess the size of the threshold contributions at $2m_\psi$. Hence, I recommend extending the plot and/or using different masses for illustration to enhance clarity in this aspect.
6) What is the rationale for setting the renormalization and factorization scales to $2m_h$ for the double Higgs simulation, as opposed to the more common practice in the literature of dynamically varying them with theoretically motivated scales?
7) What justifies the systematic uncertainty of 4% for the ZZ analysis? The referenced work (Ref. [27]) presents results in two scenarios, 8% and 4%, recognizing them as challenging targets (refer to page 9 of the same reference). Similar to Ref. [27], it would be beneficial to display the uncertainties in two scenarios: a more aggressive 4% and a more conservative 8%.
8) For the VBF channel, a 1% systematic uncertainty is specified, a choice that appears too aggressive. It is advisable to align the systematic uncertainty with the experimental analyses.
9) The manuscript lacks a conclusion section. I recommend including such a section for a more structured presentation.

---

## Round 1 · Referee Report · Christoph Englert · 2024-1-9

Report
The paper "Quantum collider probes of the fermionic Higgs portal" by Haisch and collaborators provides a summary of the sensitivity of different collider processes to a fermionic Higgs portal. The work is complete, scientifically sound, and detailed. It provides a good contribution to the ongoing BSM efforts pursued at the LHC as well as an outlook towards future colliders. I recommend publication in SciPost.

---

## Round 1 · Referee Report · Anonymous · 2024-1-16

Report
This paper studies NLO corrections to off-shell Higgs production in the ZZ channel, and to double Higgs production from a dimension-5 fermionic Higgs portal operator. It also compares the sensitivity of tree-level VBF pair production of the new fermions, through an off-shell Higgs. The studies are performed in the context of the future HL-LHC, HE-LHC and FCC colliders.
The authors are requested to provide clarifications and additional results on the following points, before the manuscript can be considered for publication:
1. The primary results are shown with a cut-off scale of f=3v. However, as the authors emphasize, the high-energy tails of the relevant kinematic distributions provide the dominant sensitivity to the new operators. Furthermore, for the HE-LHC and FCC, the authors consider invariant mass values of 1000 GeV and 1500 GeV, which are much larger than 3v. Hence, the EFT analysis is not valid in these kinematic regimes. Proper event-by-event kinematic conditions must be imposed to ensure the validity of an EFT analysis at an hadron collider.
2. In order to ensure renormalizability, the authors needed to introduce two other dim-6 SMEFT operators. But from the discussions, it seems often these operators drive the sensitivity. How is then the analysis a direct probe of the dim-5 Higgs portal operator? The results can remain the same even if the Wilson co-efficient of the dim-5 operator is zero at high scale, and those of the other two operators are non-zero.
3. Often very aggressive values of systematic uncertainties are considered. Please use values that are supported by the analyses of the experimental collaborations, and show results for at least two choices each, one conservative and another somewhat optimistic.
4. Because of the above concerns regarding the validity of the EFT analysis, the authors are requested to redo the analysis in a simple s-channel UV-completion of the dim-5 operator in Eq.(1). The simplest possibility would be to consider an SM singlet scalar mediator coupled to both the |H|^2 operator and the new fermion bi-linear. Please compute the relevant NLO corrections to the considered processes in this model, and then compare it with the EFT results by using a mapping of the EFT and simplified model parameters. In this way, we can know when the EFT analysis is actually valid and when it begins to fail. That would be an useful addition to the literature, since at the NLO level such comparisons are not often found.
While interesting t-/u-channel UV completions are also possible, they require a fermion mediator with the same electroweak charge as the SM Higgs field, thereby requiring more work in the renormalisation of the theory. These latter UV-completions will also generate additional dim-6 operators.
5. Finally, the authors are requested to compare the size of the NLO corrections from the new physics operator, with the higher order QCD and QCD-EW corrections from the SM for the studied processes. Are there any features in the kinematic distributions which distinguish these two contributions?

---

## Round 2 · Referee Report · Anonymous (Referee 1) · 2024-2-28

Report

I am satisfied with the authors' response to my initial report and I can recommend this paper for publication in SciPost.

---

## Round 2 · Referee Report · Anonymous (Referee 3) · 2024-2-29

Report

(1) In their response to the previous referee report, the authors write that most of their numerical results for hadron colliders in an EFT framework are applicable/valid in theories with a strongly-coupled UV completion, such as twin-Higgs models, for which the physical cut-off scale of the theory is Λ = 4πf. For weakly coupled UV completions such as DM models, the numerical results are very different, as the scale f needs to be chosen higher for consistent use of EFT at colliders with the given kinematic selections.

The authors also write in their response to another query raised that "Any analysis performed in an effective field theory (EFT) context requires assumptions on the UV. We work under the minimal assumption that at the cut-off scale Λ, only the dimension-five Higgs portal operator is present."

It is not clear whether these two assumptions made by the authors are consistent with each other. In particular, do the strongly-coupled UV models (such as twin-Higgs) that the authors are trying to probe by the suggested searches only generate the dimension-five Higgs portal operator at the cut-off scale Λ ? If so, please give an explicit example computation, by taking a particular twin-Higgs model, and explaining why or under which conditions only the non-zero dimension-5 Wilson co-efficient is expected to be generated at the cut-off scale in this theory, and the dimension-6 Wilson co-efficients are zero at the cut-off scale. For models in which all of these Wilson co-efficients are generated at the cut-off scale itself, the proposed searches are not a direct probe of the dimension-five Higgs portal operator, as claimed.

(2) The above two important assumptions made are not clear from the title and abstract of the paper. In particular, the abstract should clearly mention what kind of sensitivity is obtained under which assumptions (such as strongly coupled UV completion like the twin Higgs with only a dimension-5 operator generated at the cut-off scale of the theory gives a certain sensitivity, while weakly-coupled UV completions such as DM models give a different sensitivity).

---

## Round 2 · Author Response

Reply to reports on scipost_202312_00002v1

Ulrich Haisch, Maximilian Ruhdorfer, Konstantin Schmid, and Andreas Weiler, Quantum collider probes of the fermionic Higgs portal, 2311.03995.

We thank the referees for carefully reading the manuscript and their valuable comments. We try to address all the comments and suggestions in this reply and by changing the manuscript accordingly. We enclose a version of the draft to the new submission, where all modifications and additions are colored in red.

Reply to referee 1

(Q1) Please comment on the validity of the EFT analysis of Eq. 1. In particular, I observed that Eq. 26 introduces terms scaling as c_ψ^6 at the cross-sectional level for double Higgs production. It appears that these terms dominate the new physics contributions rather than the lower-dimensional ones.

(A1) Throughout our work, we assume that at the cut-off scale Λ, only the dimension-five Higgs portal operator Eq. (1) is present. Any additional effective operator is generated exclusively through renormalization group (RG) running. Under this assumption, our analysis includes all relevant contributions at the one-loop order and up to the level of dimension-six operators.

(Q2) In this context, what is the rationale behind including only the two dim-6 operators in Eq. 4? Considering that the final calculation accounts for significantly higher-order terms in the EFT expansion, what motivates the specific choice to limit the inclusion to these two operators?

(A2) As mentioned in (A1), our assumption is that at the cut-off scale, only the dimension-five Higgs portal operator has a non-vanishing Wilson coefficient. The two dimension-six operators in Eq. (4) are not a choice but are inevitably generated through RG running. At the technical level, they are the minimal set of dimension-six operators needed to make the theory ultraviolet (UV) finite at the one-loop level.

(Q3) Considering the chosen cutoff for the theory (f = 3v), there seems to be a lack of justification for the upper range of the m_4l analysis, specifically including events up to m_4 l = 1 TeV for the HE-LHC and m_4l = 1.5 TeV for the FCC.

(A3) It is important to realise that in our numerical analysis, we focus on twin-Higgs realizations of (1). Such models become strongly coupled at the scale 4πf . The UV cut-off scale is therefore Λ = 4πf ~ 9 TeV, which is safely above the relevant energy scales probed, for instance, in gg ⇾ h^* ⇾ ZZ ⇾ 4l.

(Q4) On page 11: "The inclusion of higher-order QCD corrections furthermore reduces the scale uncertainties of DS by a factor of about 3 from around 7.5% to 2.5%." Please provide more details on how these scale uncertainties are derived.

(A4) The quoted uncertainties rely on Eq. (31) in the case of the QCD-improved prediction of D_S and have been obtained from seven-point scale variations enforcing the constraint 1/2 ≤ μ_R/μ_F ≤ 2 on the renormalisation and factorisation scales μ_R and μ_F.

(Q5) Regarding Fig. 5, it would be beneficial to assess the size of the threshold contributions at 2*m_ψ. Hence, I recommend extending the plot and/or using different masses for illustration to enhance clarity in this aspect.

(A5) We find that the threshold effects are negligible and essentially invisible in the m_4l distribution.

(Q6) What is the rationale for setting the renormalization and factorization scales to 2*m_h for the double Higgs simulation, as opposed to the more common practice in the literature of dynamically varying them with theoretically motivated scales?

(A6) This choice can be motivated as follows. In the case of double-Higgs production, we are only considering the total cross-section to constrain the strength of the fermionic Higgs portal. The cross-section differential in the invariant mass of the two Higgses m_hh has a maximum at around 250 GeV = 2*m_h. Since the contributions at 2*m_h dominate the total cross-section, choosing μ_R = μ_F = 2*m_h seems well-motivated. Notice that μ_R = μ_F = m_hh/2 with m_hh evaluated dynamically is a choice commonly made in the literature on double-Higgs production.

(Q7) What justifies the systematic uncertainty of 4% for the ZZ analysis? The referenced work (Ref. [27]) presents results in two scenarios, 8% and 4%, recognizing them as challenging targets (refer to page 9 of the same reference). Similar to Ref. [27], it would be beneficial to display the uncertainties in two scenarios: a more aggressive 4% and a more conservative 8%.

(A7) Following the referee's advice, we have added an appendix, i.e. Appendix A, to our manuscript. We present results for our off-shell and VBF Higgs analyses using two different assumptions concerning the systematic uncertainties for each collider.

(Q8) For the VBF channel, a 1% systematic uncertainty is specified, a choice that appears too aggressive. It is advisable to align the systematic uncertainty with the experimental analyses.

(A8) To determine the collider reach of the VBF Higgs production channel, we now follow the methodology of the CERN Yellow report on the HL-LHC physics potential (i.e. Ref. [72]) as well as the FCC physics opportunities study (i.e. Ref. [78]). These reports give systematic uncertainties of 4% (2%) in the S1 (S2) scenario for the case of the HL-LHC, while in the case of the HE-LHC and FCC they quote 2% (1%) and 1% (0.5%), respectively.

(Q9) The manuscript lacks a conclusion section. I recommend including such a section for a more structured presentation.

(A9) We thank the referee for this suggestion. We have added a conclusion to our manuscript.

Reply to referee 3

(Q1) The primary results are shown with a cut-off scale of f = 3v. However, as the authors emphasize, the high-energy tails of the relevant kinematic distributions provide the dominant sensitivity to the new operators. Furthermore, for the HE-LHC and FCC, the authors consider invariant mass values of 1000 GeV and 1500 GeV, which are much larger than 3v. Hence, the EFT analysis is not valid in these kinematic regimes. Proper event-by-event kinematic conditions must be imposed to ensure the validity of an EFT analysis at an hadron collider.

(A1) It is important to note that the operator suppression scale f does not coincide with the UV cut-off scale Λ of the theory. The two are related by Λ = g* f, where g is the typical coupling of the theory. In theories with a strongly-coupled UV completion, such as twin-Higgs models, g is close to its maximal value of 4π. Thus 1 TeV and 1.5 TeV are well below the physical cut-off scale Λ = 4πf ~ 9 TeV. As usual, the scale f should be understood in the same way as the pion decay constant, which is of the size as the pion mass – and we can certainly reliably calculate amplitudes in chiral perturbation theory using on-shell pions.

(Q2) In order to ensure renormalizability, the authors needed to introduce two other dimension-six SMEFT operators. But from the discussions, it seems often these operators drive the sensitivity. How is then the analysis a direct probe of the dimension-five Higgs portal operator? The results can remain the same even if the Wilson coefficient of the dimension-five operator is zero at high scale, and those of the other two operators are non-zero.

(A2) Any analysis performed in an effective field theory (EFT) context requires assumptions on the UV. We work under the minimal assumption that at the cut-off scale Λ, only the dimension-five Higgs portal operator is present. The dimension-six operators are generated by RG running due to the presence of the dimension-five operator. Their Wilson coefficients are then completely determined in terms of c_ψ, making any observable sensitive to them a probe of the dimension-five Higgs portal operator under our assumptions on the UV.

(Q3) Often very aggressive values of systematic uncertainties are considered. Please use values that are supported by the analyses of the experimental collaborations, and show results for at least two choices each, one conservative and another somewhat optimistic.

(A3) Following the referee's advice, we added an appendix to our manuscript, i.e. Appendix A. We present results for our off-shell and VBF Higgs analyses using two different assumptions concerning the systematic uncertainties for each collider. To determine the collider reach of the VBF Higgs production channel, we now follow the methodology of the CERN Yellow report on the HL-LHC physics potential (i.e. Ref. [72]) as well as the FCC physics opportunities study (i.e. Ref. [78]). These reports give systematic uncertainties of 4% (2%) in the S1 (S2) scenario for the case of the HL-LHC, while in the case of the HE-LHC and FCC they quote 2% (1%) and 1% (0.5%), respectively.

(Q4) Because of the above concerns regarding the validity of the EFT analysis, the authors are requested to redo the analysis in a simple s-channel UV-completion of the dimension-five operator in Eq.(1). The simplest possibility would be to consider an SM singlet scalar mediator coupled to both the |H|^2 operator and the new fermion bi-linear. Please compute the relevant NLO corrections to the considered processes in this model, and then compare it with the EFT results by using a mapping of the EFT and simplified model parameters. In this way, we can know when the EFT analysis is actually valid and when it begins to fail. That would be an useful addition to the literature, since at the NLO level such comparisons are not often found.

(A4) While such a study would be interesting, it is not directly relevant to our work. The motivation for our analysis is twin-Higgs type models, which have a strongly-coupled UV completion with a cut-off scale Λ = 4πf — see (A1). Our analyses of off-shell Higgs production within the twin-Higgs scenarios (cf. Section 4 and Appendix B) are thus completely within the region of validity of the EFT. Notice that in the case of the DM Higgs portal model (see Appendix C), the constraints that arise from D_S are not competitive with the limits that follow from double-Higgs production. In the latter case, we only consider the total cross-section, dominated by the contributions close to the double-Higgs threshold. The signal strength μ_hh in double-Higgs production can hence be calculated with confidence using the effective Lagrangian Eq. (3) even for f = 1 TeV as chosen in Figure 13.

(Q5) Finally, the authors are requested to compare the size of the NLO corrections from the new physics operator, with the higher order QCD and QCD-EW corrections from the SM for the studied processes. Are there any features in the kinematic distributions which distinguish these two contributions?

(A5) We stress that for what concerns the D_S observable, cf. Eq. (30), we have incorporated higher-order QCD effects in both the Standard Model (SM) and the fermionic Higgs portal model using Eq. (31). The mixed QCD-EW corrections to gg ⇾ h^* ⇾ ZZ ⇾ 4l production are to the best of our knowledge not known in the SM. Therefore, we would not know how to include such effects in our study. In the case of VBF Higgs production, we also include higher-order QCD corrections in our background estimates (see p. 15). Incorporating mixed QCD-EW corrections into our background simulations may seem excessive, considering the relatively weak limits obtained from VBF Higgs production. In the case of double-Higgs production, we do not include QCD corrections. This simplification appears justified, especially considering the expected weak limits on the signal strength (μ_hh ∈ [0.7, 1.8]) the HL-LHC is anticipated to allow to set. Considering the exploratory nature of our HE-LHC and FCC double-Higgs studies, we also maintain that disregarding the impact of QCD corrections in these instances is justifiable.

Again, we thank the referees for the useful feedback and hope that the manuscript can be published in SciPost Physics in its revised form with the above explanations and the changes made.

---

## Round 3 · Author Response

We thank the referees for their additional comments. Following the advice of one of the referees, we have now studied the effect of non-zero initial conditions of the Wilson coefficients of the operators $Q_6$ and $Q_{H \Box}$ in off-shell and double-Higgs production. This study is detailed in a new Appendix D. We have also amended the abstract as recommended by this referee. During the revision process, we identified a minor mistake in our Monte Carlo implementation of double-Higgs production. We have rectified this mistake and regenerated all relevant plots, updating the text accordingly. All changes and additions are indicated in red in the resubmitted PDF file. We hope that with these revisions, our manuscript will meet the criteria for publication in SciPost.

---

## Editorial Decision

published